# Schwann cell TRPA1 mediates neuroinflammation that sustains macrophage-dependent neuropathic pain in mice

Francesco De Logu[1], Romina Nassini[1], Serena Materazzi[1], Muryel Carvalho Gonçalves[1], Daniele Nosi[2], Duccio Rossi Degl'Innocenti[1], Ilaria M. Marone[1], Juliano Ferreira[3], Simone Li Puma[1], Silvia Benemei[1], Gabriela Trevisan[4], Daniel Souza Monteiro de Araújo[1,5], Riccardo Patacchini[6], Nigel W. Bunnett[7] & Pierangelo Geppetti [1]

It is known that transient receptor potential ankyrin 1 (TRPA1) channels, expressed by nociceptors, contribute to neuropathic pain. Here we show that TRPA1 is also expressed in Schwann cells. We found that in mice with partial sciatic nerve ligation, TRPA1 silencing in nociceptors attenuated mechanical allodynia, without affecting macrophage infiltration and oxidative stress, whereas TRPA1 silencing in Schwann cells reduced both allodynia and neuroinflammation. Activation of Schwann cell TRPA1 evoked NADPH oxidase 1 (NOX1)-dependent $H_2O_2$ release, and silencing or blocking Schwann cell NOX1 attenuated nerve injury-induced macrophage infiltration, oxidative stress and allodynia. Furthermore, the NOX2-dependent oxidative burst, produced by macrophages recruited to the perineural space activated the TRPA1–NOX1 pathway in Schwann cells, but not TRPA1 in nociceptors. Schwann cell TRPA1 generates a spatially constrained gradient of oxidative stress, which maintains macrophage infiltration to the injured nerve, and sends paracrine signals to activate TRPA1 of ensheathed nociceptors to sustain mechanical allodynia.

[1] Department of Health Sciences, Section of Clinical Pharmacology and Oncology, University of Florence, Florence 50139, Italy. [2] Department of Experimental and Clinical Medicine, Section of Anatomy and Histology, University of Florence, Florence 50139, Italy. [3] Department of Pharmacology, Federal University of Santa Catarina, Florianópolis 88040-500, Brazil. [4] Laboratory of Neuropsychopharmacology and Neurotoxicity, Graduate Program in Pharmacology, Federal University of Santa Maria (UFSM), Santa Maria 97105-900, Brazil. [5] Department of Neurobiology and Program of Neurosciences, Institute of Biology, Fluminense Federal University, Niterói, 20010-060, Brazil. [6] Department of Pharmacology, Chiesi Farmaceutici SpA, Parma 43122, Italy. [7] Departments of Surgery and Pharmacology, Columbia University, New York, NY 10027, USA. Francesco De Logu and Romina Nassini contributed equally to this work. Correspondence and requests for materials should be addressed to P.G. (email: geppetti@unifi.it)

Neuropathic pain, which is defined as pain caused by a lesion or disease of the somatosensory nervous system[1], encompasses a large variety of conditions[2]. Lesions of the peripheral nervous system can cause lifelong neuropathic pain. Following peripheral nerve injury, local infiltration of inflammatory cells, a hallmark of Wallerian degeneration, occurs[3–5], and is associated with the development of neuropathic pain. Although the infiltration of macrophages into the damaged nerve trunk is known to induce mechanical allodynia in mice with sciatic nerve injury[6–9], the precise pathway by which inflammatory cells cause persistent allodynia is only partially defined. A series of mediators have been reported to contribute to macrophage infiltration in the damaged nerve[10]. Notably, inhibition of the chemokine (C–C motif) ligand 2 (CCL2) has been shown to attenuate neuroinflammation and allodynia[7,8,11]. Oxidative stress contributes to neuropathic pain, since antioxidants attenuate mechanical hypersensitivity in mouse models, including chronic constriction of the sciatic nerve[12] and spinal nerve ligation[13].

The transient receptor potential ankyrin 1 (TRPA1) channel is highly expressed by a subpopulation of primary sensory neurons[14,15] that contain and release the proinflammatory neuropeptides substance P (SP) and calcitonin gene-related peptide (CGRP)[15]. TRPA1 is activated by a series of exogenous agents, including allyl isothiocyanate (AITC)[16,17], and is typically sensitive to the redox state of the milieu[18]. Notably, a series of reactive oxygen, nitrogen or carbonyl species, including hydrogen peroxide ($H_2O_2$), activate TRPA1, resulting in nociceptor stimulation or sensitization[19–24]. TRPA1 has been shown to mediate mechanical hypersensitivity in different models of inflammatory and neuropathic pain, including those evoked by peripheral nerve injury[25–29].

Recent findings in mice with trigeminal nerve injury (constriction of the infraorbital nerve, CION) show that macrophages, recruited by a CCL2-dependent process, increase $H_2O_2$ levels within the site of nerve injury[30]. The resulting oxidative stress and the ensuing increases in reactive carbonyl species were proposed to mediate prolonged mechanical allodynia by gating TRPA1 in trigeminal nerve fibers[30]. Thus, TRPA1, expressed by primary sensory neurons, appears to be the target of the macrophage-dependent oxidative burst required to promote neuropathic pain. Here, we surprisingly found that pharmacological blockade or genetic deletion of TRPA1 not only induced the expected inhibition of mechanical allodynia, but also suppressed macrophage infiltration and $H_2O_2$ generation in the injured nerve. The current study was undertaken to identify the cellular and molecular mechanisms responsible for this TRPA1-mediated macrophage infiltration and generation of oxidative stress.

By using pharmacological and genetic approaches to disrupt TRPA1, including conditional deletion in Schwann cells, we found that Schwann cells that ensheath the injured sciatic nerve axons express TRPA1. Macrophages, which are recruited by CCL2, generate a NADPH oxidase-2 (NOX2)-dependent oxidative burst that targets Schwann cell TRPA1. TRPA1, via NOX1, produces sustained oxidative stress that maintains, in a spatially confined manner, macrophage infiltration into the injured nerve, and which activates TRPA1 on nociceptor nerve fibers to produce allodynia.

## Results

### TRPA1 mediates neuroinflammation.
In C57BL/6 mice pSNL, but not sham surgery (Fig. 1a), induced prolonged (3–20 days) mechanical allodynia (Fig. 1b) accompanied by macrophage (F4/80+ cells) recruitment (Fig. 1c, e and Supplementary Fig. 1) and oxidative stress ($H_2O_2$) generation (Fig. 1d) within the injured nerve. Trpa1 (Fig. 1f), but not Trpv1 or Trpv4 (Supplementary

Fig. 2a), deletion prevented mechanical allodynia. Trpa1, but not Trpv1 or Trpv4, deletion also attenuated cold allodynia, but this response was not further investigated in the present study (Supplementary Fig. 2b). Heat hyperalgesia was unaffected by Trpa1, Trpv1, and Trpv4 deletion (Supplementary Fig. 2c). As previously reported[28,30,31] in similar models, at day 10 after pSNL (all measurements were at 10 days unless otherwise specified), TRPA1 antagonists (HC-030031, A-967079) and antioxidants (α-lipoic acid (αLA) and phenyl-N-tert-butylnitrone (PBN)) (Fig. 1g and Supplementary Fig. 3a) reversed mechanical allodynia. Treatments for 3 days with the monocyte-depleting agent clodronate[32] or an anti-CCL2 antibody (CCL2-Ab)[30,33] attenuated allodynia, macrophage infiltration and $H_2O_2$ generation (Supplementary Fig. 4a–c), confirming the proalgesic role of these cells.

Other inflammatory cells, which are recruited to sites of nerve injury, may also contribute to mechanical allodynia[9,34]. To explore their role in the delayed phase of mechanical allodynia, the number of neutrophils and T lymphocytes was evaluated in the nerve trunk at day 10 after surgery. Although both neutrophils (Ly6g+ cells) and T lymphocytes (CD8+ cells) were increased by pSNL (Supplementary Fig. 4d), treatment with clodronate, which markedly attenuated both the infiltrating macrophages and allodynia, did not affect the number of neutrophils or T lymphocytes (Supplementary Fig. 4d). In agreement with a previous report[34], these data exclude the contribution of neutrophils and T cells to mechanical allodynia assessed 10 days after pSNL.

The hypothesis that oxidative stress produced by infiltrating macrophages targets neuronal TRPA1 to signal neuropathic pain[30] implies that the channel inhibition reduces allodynia but does not affect neuroinflammation. Surprisingly, Trpa1 deletion prevented infiltration of F4/80+ cells and $H_2O_2$ generation in the injured sciatic nerve (Fig. 1h, i). TRPA1 antagonists (Fig. 1h, j, k) and antioxidants (Fig. 1h, l and Supplementary Fig. 3b) also transiently reversed macrophage infiltration and $H_2O_2$ production. Thus, the TRPA1-oxidative stress pathway mediates both neuropathic pain and neuroinflammation in the injured nerve.

### CCL2 induces neuroinflammation via TRPA1.
One possible explanation may be that TRPA1 mediates the release of the monocyte chemoattractant, CCL2, generated by injured nerves[8]. However, as neither TRPA1 deletion or antagonism nor antioxidants affected CCL2 increases in ligated sciatic nerves (Fig. 2a), the chemokine should originate from a TRPA1-oxidative stress-independent pathway. As previously shown[8,35], local perineural CCL2 administration induced mechanical allodynia, as well as producing F4/80+ cell infiltration and $H_2O_2$ generation (Fig. 2b, c). TRPA1 deletion or antagonism and antioxidants prevented or reversed the effects of CCL2 (Fig. 2b, c). Pretreatment with clodronate, which depletes circulating monocytes and thereby inhibits their neural accumulation[30], prevented mechanical allodynia evoked by CCL2 (Supplementary Fig. 4e). Furthermore, in mice with pSNL clodronate treatment depleted macrophages and attenuated mechanical allodynia (Supplementary Fig. 4a), but did not affect the increased CCL2 levels within the ligated nerve trunk (Supplementary Fig. 4f). Together, the present findings support the view that oxidative stress and TRPA1 induce neuroinflammation downstream from CCL2. There was a distinct temporal difference between the effects of CCL2-Ab and TRPA1 antagonists/antioxidants on pSNL-induced neuroinflammation and allodynia. One-hour after HC-030031, A-967079, αLA or PBN, pSNL-induced F4/80+ cell infiltration, $H_2O_2$ formation and allodynia were all prominently inhibited (Fig. 1g, h, j–l and Supplementary

Fig. 3b), whereas a high dose of the CCL2-Ab, which at 1 h already attenuated neural CCL2 levels, was completely ineffective (over 6 h) in reducing mechanical allodynia (Fig. 2d). Successful inhibition of pain and inflammation required the administration of a lower CCL2-Ab dose for 3 consecutive days that, as

expected[30,33], also reduced CCL2 levels in the nerve trunk (Supplementary Fig. 4b). Thus, while TRPA1-antagonism/anti-oxidants rapidly (within 1 h) reversed neuroinflammation, CCL2-blockade required a much longer time (3 days) to produce the same inhibitory effects[30,33].

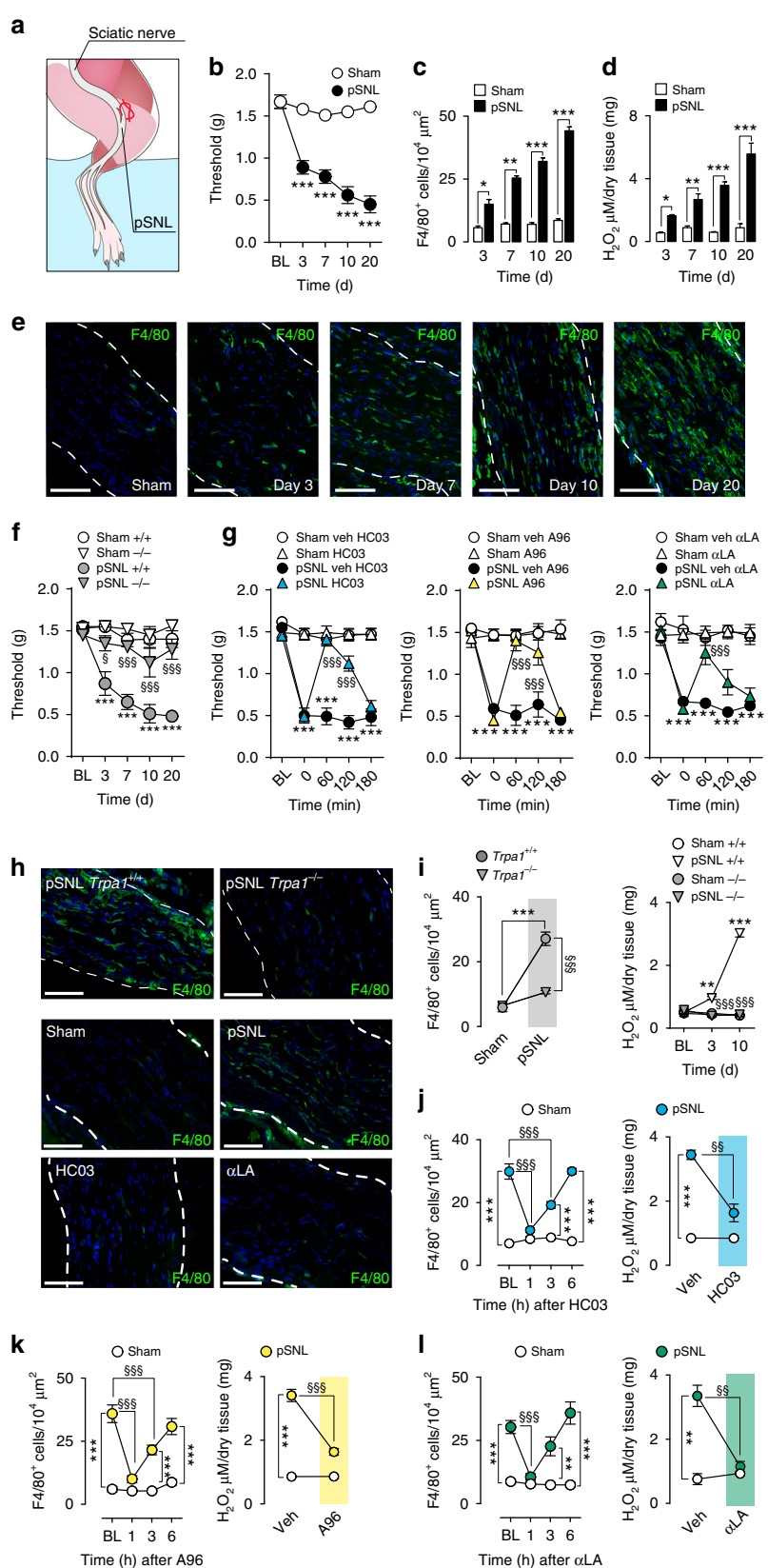

**Schwann cells express TRPA1 and release $H_2O_2$.** Nerve fibers, macrophages and Schwann cells within the injured nerve trunk could potentially mediate TRPA1-dependent oxidative stress. By targeting TRPV1, resiniferatoxin (RTX) defunctionalizes TRPV1$^+$/TRPA1$^+$ neurons[36–38]. RTX abolished nociceptive responses to TRPV1 (capsaicin) and AITC (Supplementary Fig. 5a) and reversed pSNL-evoked allodynia, but did not affect F4/80$^+$ cell infiltration and $H_2O_2$ generation (Supplementary Fig. 5b). Thus, TRPV1$^+$/TRPA1$^+$ nerve fibers mediate neuropathic pain but not neuroinflammation. Naïve or lipopolysaccharide-activated mouse peritoneal macrophages in culture neither expressed TRPA1 mRNA (RT-qPCR), TRPA1 protein (immunocytochemistry) nor responded to AITC (Ca$^{2+}$-signaling) (Supplementary Fig. 5c–e). Thus, macrophages infiltrating the injured nerve cannot generate TRPA1-dependent oxidative stress.

Schwann cells ensheath nerve fibers, including C-fiber nociceptors, and represent 90% of the nucleated cells of the nerve trunk[39]. We localized immunoreactive TRPA1 to PGP9.5$^+$ nerve fibers and to S-100$^+$ or SOX10$^+$ Schwann cells in the sciatic nerve trunk from C57BL/6 mice (Fig. 3a–c). TRPA1 immunoreactivity was not detected in dorsal root ganglia (DRG, L4-L6) (Fig. 3d) or in S-100$^+$ cells in the sciatic nerve trunk (Fig. 3e) from Trpa1$^{-/-}$ mice, which confirms antibody selectivity. Expression of TRPA1 (protein and mRNA) in cultured mouse Schwann cells was confirmed by immunofluorescence, western blotting and RT-qPCR (Fig. 3f–h and Supplementary Fig. 6a). Furthermore, AITC-induced intracellular Ca$^{2+}$ response in cultured Schwann cells from wild type mice, which was attenuated by HC-030031 (Fig. 4a). In contrast, capsaicin or a TRPV4 agonist (GSK1016790A) failed to produce any Ca$^{2+}$ response (Fig. 4a). Importantly, Schwann cells from Trpa1$^{+/+}$ mice, but not from Trpa1$^{-/-}$ mice, responded to AITC (Fig. 4b). The ability of TRPA1 to promote oxidative stress was explored by measuring $H_2O_2$ generation. Both AITC and $H_2O_2$, which has been shown to gate TRPA1[20], stimulated $H_2O_2$ release from HEK293 cells expressing the human TRPA1 (hTRPA1-HEK293 cells), but not from untransfected HEK293 cells (Fig. 4c, d). AITC or $H_2O_2$ also produced a time- and Ca$^{2+}$-dependent $H_2O_2$ generation in Schwann cells, which was prevented by HC-030031 (Fig. 4e). Thus, Schwann cells generate $H_2O_2$ in response to TRPA1 activation.

**Schwann cells generate oxidative stress via TRPA1 and NOX1.** Different NOXs play a key role in oxidative stress[40]. While we observed that macrophages contain exclusively NOX2 (mRNA and protein) (Supplementary Fig. 5c, f, g), cultured Schwann cells expressed NOX1, NOX2 and NOX4 mRNAs (Supplementary Fig. 6a). Yet, only NOX1 protein was detected by immunofluorescence (Fig. 5a and Supplementary Fig. 6b). Functional data corroborated the molecular/morphological findings as the selective NOX2-inhibitor (Gp91ds-tat peptide)[41], but not the NOX1/

NOX4 inhibitor (GKT13783)[42], attenuated phorbol myristate acetate-stimulated $H_2O_2$ release from cultured peritoneal macrophages (Supplementary Fig. 5h). However, GKT13783, but not gp91ds-tat peptide, inhibited AITC-induced $H_2O_2$ release from cultured Schwann cells (Supplementary Fig. 5i), indicating a major role for NOX1. 1 h after the administration of 2-acetylphenothiazine (ML171, selective NOX1-inhibitor)[43] or GKT13783, pSNL-induced allodynia and neuroinflammation were attenuated (Fig. 5b and Supplementary Fig. 6c), whereas gp91ds-tat peptide was ineffective (Supplementary Fig. 6d). Perineural administration of antisense oligonucleotides (AS-ODN) for NOX1, NOX2 or NOX4 effectively knocked down respective mRNA expression (Fig. 5c and Supplementary Fig. 6e, g). However, only NOX1 AS-ODN attenuated pSNL-induced allodynia, macrophage infiltration and $H_2O_2$ generation (Fig. 5d and Supplementary Fig. 6f, h).

To define the contribution of the TRPA1/NOX1 pathway of Schwann cells to neuroinflammation and neuropathic pain evoked by pSNL, antisense TRPA1 oligonucleotide (TRPA1 AS-ODN) or their mismatched (TRPA1 MM-ODN) analogs were administered to pSNL mice by perineural or intrathecal route (Fig. 6a, f). Perineural treatment with TRPA1 AS-ODN did not affect expression of TRPA1 mRNA and immunoreactivity in DRGs (L4-L6), or acute nociceptive responses to perineural AITC or capsaicin (Fig. 6b). However, it markedly reduced TRPA1 mRNA and the TRPA1/S-100 overlap (Fig. 6c) in injured sciatic nerves, and inhibited pSNL-evoked allodynia, neural F4/80$^+$ cell infiltration, and $H_2O_2$ generation (Fig. 6d, e). Thus, Schwann cell TRPA1 silencing preserved TRPA1-mediated acute nociceptive signaling, but disrupted pSNL-evoked neuroinflammation and allodynia. Intrathecal administration of TRPA1 AS-ODN (Fig. 6f) down-regulated TRPA1 mRNA and immunoreactivity[44] in DRGs (L4-L6), suppressed the acute nociception by perineural AITC, but not capsaicin, and inhibited pSNL-evoked allodynia (Fig. 6g, i). However, TRPA1 mRNA, the TRPA1/S-100 overlap, and pSNL-evoked F4/80$^+$ cell infiltration and $H_2O_2$ generation in injured nerves were unaffected (Fig. 6h, j).

HC-030031 reduced the number of fluorescent macrophages accumulated at the site of pSNL by ~50% (Fig. 7a), assessed by in vivo imaging[45]. Increased F4/80$^+$ cells were found in the tissue surrounding the injured nerve trunk (Fig. 7b). HC-030031 or αLA reduced F4/80$^+$ cells in the pSNL-injured nerve trunk and neighboring tissue, but not in perineural tissue distant from the injury site (Fig. 7b). The Schwann cell-mediated TRPA1/NOX1 pathway regulates the final stages of macrophage migration from the circulation into the injured nerve trunk, in a manner dependent on the $H_2O_2$ concentration gradient.

To provide further support for the involvement of Schwann cell TRPA1 in orchestrating neuroinflammation and ensuing neuropathic pain in the pSNL model, we selectively deleted TRPA1 from Schwann cells. We crossed a floxed TRPA1 mouse

---

**Fig. 1** TRPA1 mediates pSNL-evoked allodynia and neuroinflammation. **a** Drawing representing the pSNL surgery in mice. **b–e** Time-dependent (3–20 days, d) mechanical allodynia (**b**), number and representative images of macrophages (F4/80$^+$ cells) (**c, e**) and $H_2O_2$ content (**d**) in the sciatic nerve trunk induced by pSNL in C57BL/6 compared to sham mice ($n = 6$, $*P < 0.005$, $**P < 0.01$ $***P < 0.001$ pSNL vs. Sham; two-way ANOVA followed by Bonferroni post hoc analyses and unpaired two-tailed Student's t-test). **f** Time-dependent (3–20 d) mechanical allodynia in sham/pSNL Trpa1$^{+/+}$/Trpa1$^{-/-}$ mice ($n = 8$, $***P < 0.001$ pSNL$^{+/+}$ vs. Sham$^{+/+}$; $n = 6$, $§P < 0.05$ and $§§§P < 0.001$ pSNL$^{-/-}$ vs. pSNL$^{+/+}$; two-way ANOVA followed by Bonferroni post hoc analyses). **g** Mechanical allodynia (at day 10 after surgery) in sham/pSNL mice after HC-030031 (HC03, 100 mg kg$^{-1}$, i.p.), A-967079 (A96, 100 mg/kg, i.p.) and α-lipoic acid (αLA, 100 mg kg$^{-1}$, i.p.) or respective vehicles (veh, 4% DMSO and 4% tween 80 in isotonic saline) ($n = 6$, $***P < 0.001$ pSNL veh vs. Sham veh; $§§§P < 0.001$ pSNL-HC03, A96 or αLA vs. pSNL-veh; two-way ANOVA followed by Bonferroni post hoc analyses). **h–l** Representative images, number of F4/80$^+$ cells, and $H_2O_2$ content in the sciatic nerve of sham/pSNL Trpa1$^{+/+}$/Trpa1$^{-/-}$ and C57BL/6 mice, before (BL) and 1–6 h after HC03, A96, αLA (all, 100 mg kg$^{-1}$, i.p.) or respective vehicles (veh, 4% DMSO and 4% tween 80 in isotonic saline) ($n = 6$, $**P < 0.01$ and $***P < 0.001$ pSNL Trpa1$^{+/+}$ vs. Sham-Trpa1$^{+/+}$ and pSNL veh vs. Sham veh; $§§P < 0.01$ and $§§§P < 0.001$ pSNL Trpa1$^{-/-}$ vs. pSNL-Trpa1$^{+/+}$ and pSNL HC03, A96 or αLA vs. pSNL-veh; two-way and one-way ANOVA followed by Bonferroni post hoc analyses). (Scale bars: 50 μm; (**e, h**) dashed lines, perineurium). Data are represented as mean ± s.e.m

(TRPA1fl/fl) with a *Plp1-Cre/ERT* mouse in which Cre recombinase is expressed in Schwann cells/oligodendrocytes (*Plp1-Cre$^{ERT}$;Trpa1$^{fl/fl}$* mice). *Plp1-Cre$^{ERT}$;Trpa1$^{fl/fl}$* mice were treated with tamoxifen to induce TRPA1 deletion in Schwann cells. In *Plp1-Cre$^{ERT}$;Trpa1$^{fl/fl}$* mice, the ability of intraplantar injection of AITC to evoke acute nociception was unaffected (Fig. 8a). In nerve trunks from *Plp1-Cre$^{ERT}$;Trpa1$^{fl/fl}$* mice, immunoreactive TRPA1 was detected in PGP9.5$^+$ nerve fibers, indicating preservation of TRPA1 expression in sensory nerve fibers

(Fig. 8b). In contrast, immunoreactive TRPA1 was markedly down-regulated in S100$^+$ cells, but not in PGP9.5$^+$ nerve fibers, confirming effective channel deletion in Schwann cells (Fig. 8b). Functional confirmation of the selective conditional *Trpa1* gene knock-out was obtained by the failure of AITC to increase $[Ca^{2+}]_i$ in Schwann cells from *Plp1-Cre$^{ERT}$;Trpa1$^{fl/fl}$* mice (Fig. 8c). In *Plp1-Cre$^{ERT}$;Trpa1$^{fl/fl}$* mice, mechanical allodynia (Fig. 8d) and macrophage recruitment and oxidative stress (H$_2$O$_2$ generation) in the injured nerve (Fig. 8e) were markedly attenuated. Thus,

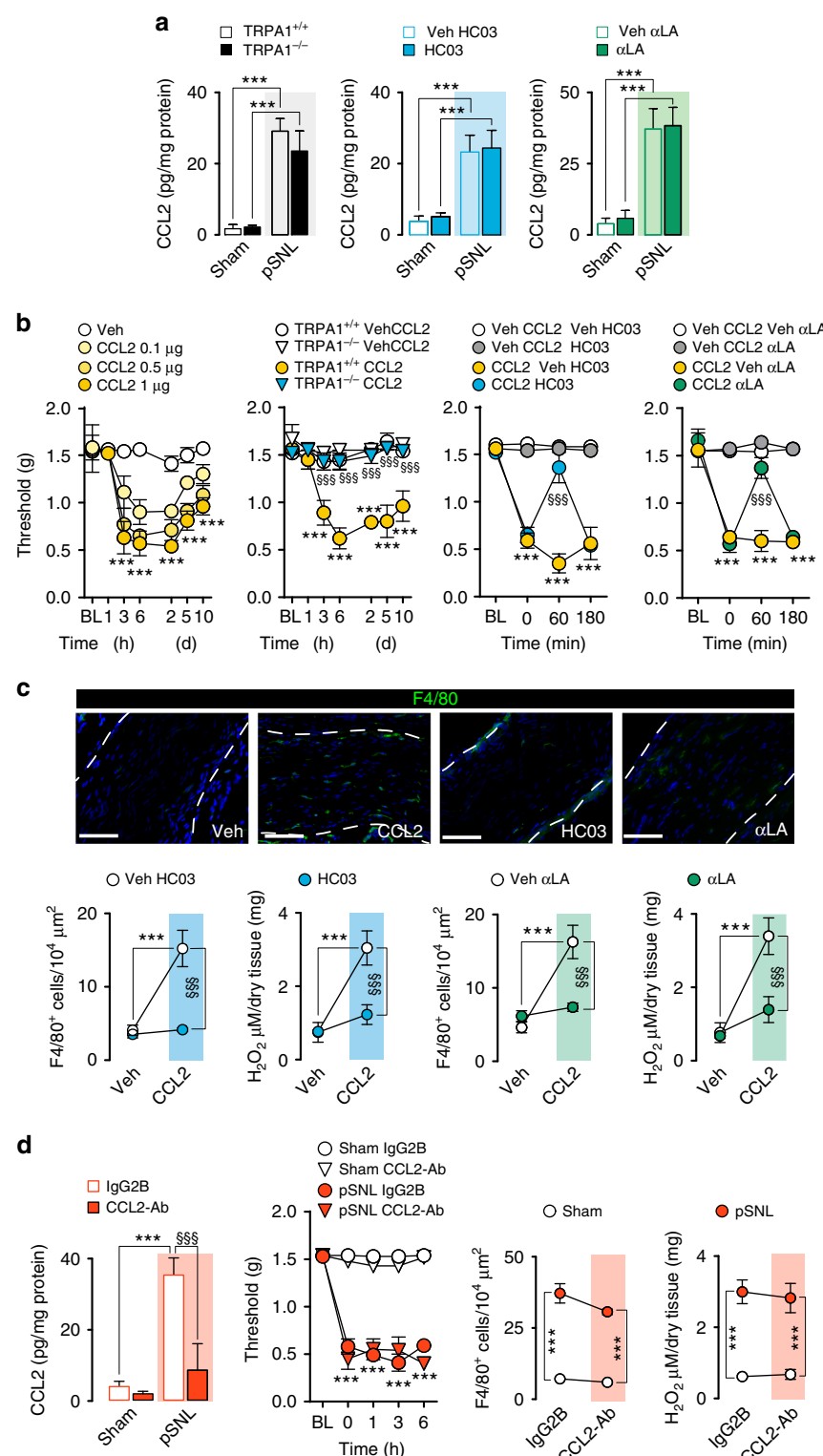

tissue-selective deletion shows that Schwann cell TRPA1 promotes macrophage infiltration and oxidative stress in injured nerve trunks, whereas nociceptor TRPA1 does not contribute to neuroinflammation.

## Discussion

We report the discovery of a role for TRPA1 in Schwann cells in neuroinflammation and ensuing neuropathic pain. Previous studies have implicated TRPA1 in primary sensory neurons as a mediator of mechanical allodynia[14,15,46]. The established capacity of TRPA1 to sense oxidative stress[15,18,19,22], and recent data obtained in a model of neuropathic pain caused by trigeminal nerve injury[30], led to the hypothesis that mechanical allodynia is sustained by the oxidative burden generated by infiltrating macrophages that continuously target TRPA1 in nerve fibers. Our present results support the view that nociceptor TRPA1 is the ultimate peripheral target to signal pSNL-evoked allodynia to the brain. However, our findings demonstrate that Schwann cell TRPA1, rather than neuronal TRPA1, orchestrates the neuroinflammation and oxidative stress that sustain neuropathic pain (Fig. 9).

Diverse lines of evidence support the hypothesis that Schwann cell TRPA1 is necessary and sufficient to mediate neuroinflammation and neuropathic pain. TRPA1 blockade, achieved with chemically unrelated antagonists, markedly decreased macrophage accumulation and the oxidative burden in the injured nerve. Studies of $Trpa1^{-/-}$ mice confirmed the findings obtained with pharmacological antagonists. Although in the present model of neuropathic pain both approaches unequivocally demonstrated the key role of TRPA1, they could not discriminate between the specific contribution of neuronal vs. non-neuronal channels. TRPA1 is expressed by peptidergic primary sensory neurons that, by releasing SP and CGRP, promote neurogenic inflammation[47–49]. Stimulants of neurogenic inflammation, including the prototypic TRPV1 agonist capsaicin, evoke a transient and moderate inflammatory response, which is chemokine/cytokine-independent and is characterized by CGRP-mediated arteriole vasodilatation and SP-mediated plasma protein and leukocyte extravasation from postcapillary venules[15,50]. Although neurogenic inflammation neither induces the infiltration of inflammatory cells after nerve injury nor mediate neuropathic pain, it does contribute to migraine attacks[51,52]. In contrast, neuroinflammation is a localized and persistent inflammatory process that is confined to the injured nerve and neighboring tissues. The hallmarks of neuroinflammation encompasses chronic infiltration of leukocytes, activation of glial cells, and increased production of inflammatory mediators, including a series of cytokines and chemokines and neuropathic pain[5,53]. Experiments with RTX, which defunctionalizes TRPV1/TRPA1-expressing neurons and abrogate their sensory and proinflammatory efferent functions[36,54,55], exclude the possibility that TRPA1-dependent neurogenic inflammation contributes to pSNL-evoked neuroinflammation. RTX attenuated mechanical allodynia, but not macrophage number or $H_2O_2$ levels, which suggests that TRPA1 present in $TRPV1^+$ peptidergic neurons may signal allodynia, but does not promote the neuroinflammatory component.

We observed that the site-specific (perineural vs. intrathecal) administration of TRPA1 AS-ODN efficiently disrupted TRPA1 expressed in nociceptors or Schwann cells, respectively, as demonstrated by behavioral and molecular studies. A reduced expression of the nociceptor TRPA1 was associated with attenuation of pain, whereas diminished expression of Schwann cell TRPA1 inhibited both pain and neuroinflammation. These findings support the hypothesis that non-neuronal TRPA1 channels exert a key role in inflammatory cell recruitment and oxidative stress generation. Confirmation of this proposal was derived from experiments with $Plp1\text{-}Cre^{ERT};Trpa1^{fl/fl}$ mice, which exhibited selective depletion of $Trpa1$ in Schwann cells and markedly attenuated neuroinflammation and mechanical allodynia. This localization of TRPA1 in Schwann cells represents a plausible explanation for the widely-reported efficacy of TRPA1 antagonists in different models of neuropathic pain produced by nerve injury[25,27,28,30], where neuroinflammation is the underlying mechanism of the ongoing pain condition.

The CCL2 receptor (CCR2) is expressed by primary sensory neurons[56,57], and CCL2 has been shown to increase TRPV1 expression[58] and to sensitize TRPA1 and TRPV1[59] in nociceptors. CCL2 is upregulated during neuronal injury, and may activate its cognate receptor CCR2 on TRPV1-positive nociceptors[58]. The CCL2 system has been reported to augment nociceptor sensitivity by increasing TRPV1 expression[58] and TRPA1 and TRPV1 function[59]. The present findings, showing that CCL2 rapidly increases neuronal hypersensitivity, support the view that this chemokine may directly stimulate primary sensory neurons, thereby enhancing mechanical allodynia under short-lived experimental conditions[59,60]. However, as indicated by studies with macrophage depletion, CCL2 requires the contribution of infiltrated macrophages within the injured nerve trunk to sustain the allodynia in a prolonged model of neuropathic pain, such as the pSNL in mice. Neutrophils and lymphocytes have been reported to accumulate, although at a minor extent compared to macrophages, at sites of nerve damage, where they may contribute to the initial[9], but not delayed phase[34], of neuropathic pain. Their role in mechanical allodynia at day 10 after surgery is further excluded by the present observation that clodronate attenuated allodynia and macrophage infiltration, whereas the influx of neutrophils and lymphocytes was unchanged.

Our results reveal distinct kinetics of macrophage accumulation by CCL2 and the TRPA1/oxidative stress pathways. Despite

**Fig. 2** TRPA1 mediates CCL2-evoked allodynia and neuroinflammation. **a** CCL2 levels in sciatic nerves (at day 10 after surgery) of sham/pSNL $Trpa1^{+/+}/Trpa1^{-/-}$ and C57BL/6 mice after HC-030031 (HC03, 100 mg kg$^{-1}$, i.p.), α-lipoic acid (αLA, 100 mg kg$^{-1}$, i.p.) or respective vehicles (veh, 4% DMSO and 4% tween 80 in isotonic saline) ($n = 6$, ***$P < 0.001$ pSNL-$Trpa1^{+/+}$ vs. sham-$Trpa1^{+/+}$ and pSNL-veh vs. sham-veh; one-way ANOVA followed by Bonferroni post hoc analyses). **b** Mechanical allodynia induced by perineural CCL2 (0.1–1 μg) or vehicle (veh, isotonic saline) in C57BL/6 mice ($n = 4$, ***$P < 0.001$ veh vs. CCL2 (1 μg), two-way ANOVA followed by Bonferroni post hoc analyses) and CCL2 (1 μg) in $Trpa1^{+/+}/Trpa1^{-/-}$ and after HC03, αLA (both, 100 mg/kg, i.p.) or respective vehicles (veh, 4% DMSO and 4% tween 80 in isotonic saline) in C57BL/6 mice ($n = 4$, ***$P < 0.001$ $Trpa1^{+/+}$ CCL2 vs. $Trpa1^{+/+}$veh; CCL2 veh HC03, αLA vs. veh CCL2; §§§$P < 0.001$ $Trpa1^{-/-}$ CCL2 vs. $Trpa1^{+/+}$ CCL2 and CCL2 HC03, αLA vs. CCL2 veh HC03, αLA; two-way ANOVA followed by Bonferroni post hoc analyses). **c** Representative images, F4/80$^+$ cell number and $H_2O_2$ content in sciatic nerves of mice treated with perineural CCL2 (1 μg) after HC03, αLA (both, 100 mg kg$^{-1}$, i.p.) or respective vehicles (veh, 4% DMSO and 4% tween 80 in isotonic saline) ($n = 5$, ***$P < 0.001$ CCL2 vs. veh HC03, αLA; §§§$P < 0.001$ CCL2 HC03, αLA vs. CCL2 veh HC03, αLA; one-way ANOVA followed by Bonferroni post hoc analyses) (Scale bars: 50 μm, dashed lines indicate perineurium). **d** CCL2 levels, mechanical allodynia, F4/80$^+$ cell number and $H_2O_2$ content in sciatic nerves (at day 10 after surgery) of sham/pSNL C57BL/6 mice after an anti-CCL2 antibody (CCL2-Ab) or IgG2B control (120 μg 200 μl$^{-1}$, i.p., single administration) ($n = 6$, ***$P < 0.001$ pSNL-CCL2-Ab vs. sham-CCL2-Ab; §§§$P < 0.001$ pSNL-CCL2-Ab vs. pSNL-IgG2B; one-way ANOVA followed by Bonferroni post hoc analyses). Data are represented as mean ± s.e.m

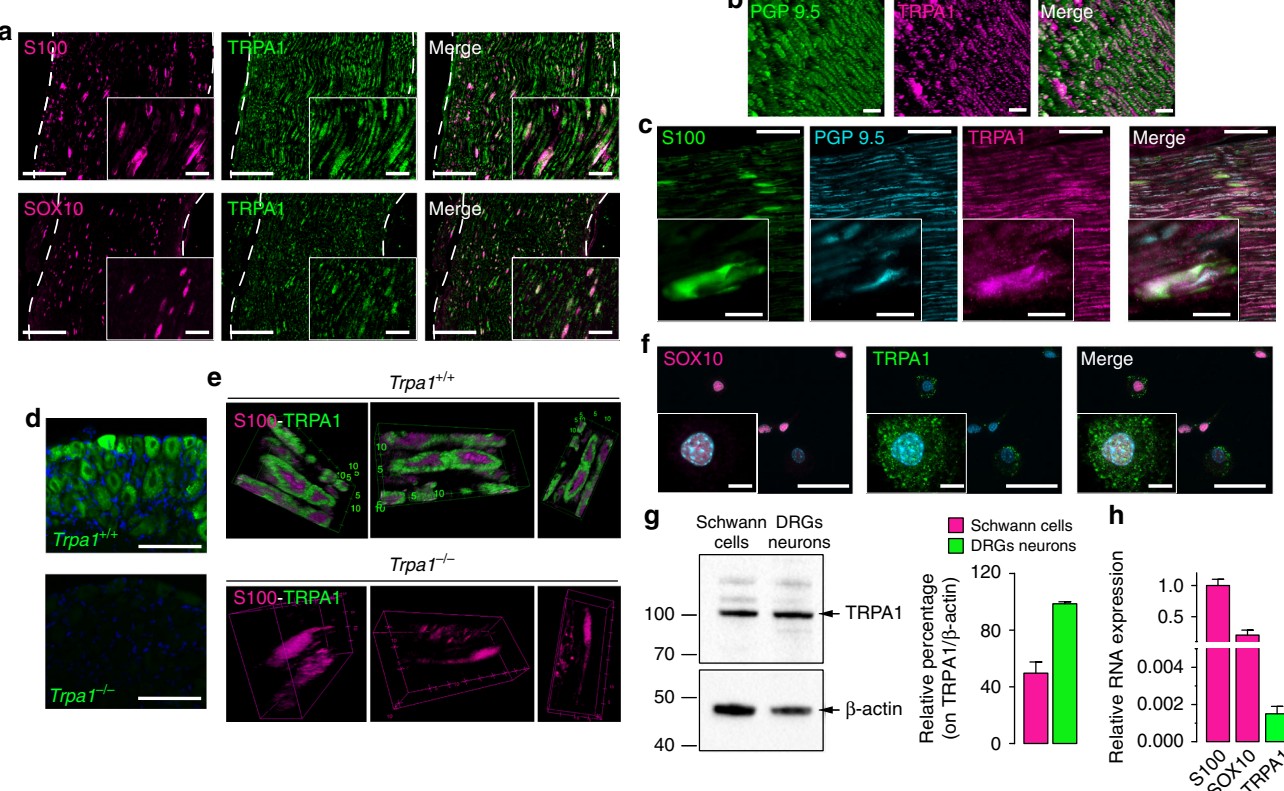

**Fig. 3** Schwann cells express TRPA1. **a** Double immunofluorescence staining of TRPA1 and S-100 and SOX10 (two specific markers for detecting Schwann cells), in sciatic nerve from C57BL/6 mice (Scale bars: 50 µm and inset 20 µm). **b** 3D confocal images of TRPA1 and PGP9.5 staining in Schwann cells from sciatic nerve trunks of C57BL/6 mice (Scale bars: 20 µm). **c** Triple immunofluorescence staining of S-100, PGP9.5 and TRPA1 in sciatic nerve trunks from C57BL/6 mice (Scale bars: 20 µm and inset 10 µm). **d** TRPA1 staining in DRGs neurons from $Trpa1^{+/+}$ and $Trpa1^{-/-}$ mice (Scale bars: 50 µm). **e** 3D confocal image reconstructions of TRPA1 and S100 in Schwann cells from sciatic nerve trunks of $Trpa1^{+/+}$ and $Trpa1^{-/-}$ mice. **f** TRPA1 and SOX-10 immunoreactivity in cultured C57BL/6 mouse Schwann cells (Scale bars: 50 µm and inset 10 µm). **g** Representative blot and TRPA1 protein content in cultured Schwann cells and DRGs neurons taken from C57BL/6 mice. Equally loaded protein was checked by expression of β-actin ($n = 4$ independent experiments). **h** TRPA1 mRNA relative expression in cultured C57BL/6 mouse Schwann cells ($n = 3$ replicates from two independent experiments). Data are represented as mean ± s.e.m

its ability to suppress perineural CCL2 levels within 1 h, a high dose of the CCL2-Ab failed to affect macrophage number, oxidative stress, and allodynia over 6 h. In contrast, a prolonged, 3-day antibody treatment was required for successful inhibition of neuroinflammation and pain[30]. The persistent temporal frame necessary for CCL2 inhibition to attenuate neuroinflammation and pain is, therefore, markedly different from the very short time-period (1–3 h) required by TRPA1 antagonists or anti-oxidants to produce the same inhibitory responses. Oxidative burst has been reported to exert a chemoattractant activity toward macrophages[61], which is limited by time and spatial constrains. Leukocyte-induced $H_2O_2$ release is a rapid event, lasting a few seconds[62], and is spatially confined to a range that does not exceed a few hundred µm[63] (Fig. 7b). Our data, including those obtained by genetic or pharmacological manipulation of NOXs, are consistent with previous observations. Macrophages express solely NOX2[40], while Schwann cells, which potentially express mRNAs for NOX1, NOX2, and NOX4, apparently express only the NOX1 protein. Since NOX1, but not NOX2 or NOX4, inhibitors or AS-ODNs attenuated neuroinflammation and allodynia, it is possible to propose that Schwann cell TRPA1 activates intracellular pathways, including $Ca^{2+}$ transients, resulting in NOX1-dependent release of oxidant molecules. Furthermore, the prominent role of NOX1, but not of NOX2, in generating allodynia excludes phagocyte-derived oxidative burst in the final activation of nociceptor TRPA1.

The most parsimonious explanation of the present results is that oxidative stress generated by Schwann cell TRPA1/NOX1 has bidirectional effects. The inwardly released $H_2O_2$ targets TRPA1 on adjacent nociceptor nerve fibers in a paracrine fashion to sustain allodynia. The outwardly released $H_2O_2$ promotes the final part (about 200 µm) of the journey of macrophages, which, deriving from the blood stream, slowly accumulate into the perineural space following the CCL2 gradient. Thereafter, following the Schwann cell-derived oxidative stress gradient, macrophages rapidly pass across the perineurium to enter the damaged nerve trunk (Fig. 9). TRPA1 has been identified in oligodendrocytes, with possible detrimental roles in ischemia and neurodegeneration[64]. Herein, we extend this observation to Schwann cells, the peripheral analogs of oligodendrocytes, which, via TRPA1, orchestrate neuroinflammation and ensuing neuropathic pain. Amelioration of neuropathic pain by currently developed TRPA1 antagonists may derive from their ability to attenuate macrophage-dependent neuroinflammation.

## Methods

**Animals and drugs**. *In vivo* experiments and tissue collection were carried out according to the European Union (EU) guidelines for animal care procedures and the Italian legislation (DLgs 26/2014) application of the EU Directive 2010/63/EU. Studies were conducted under University of Florence research permits #204/2012-B and #194/2015-PR. C57BL/6 mice (male, 20–25 g, 5–6 weeks; Envigo, Milan, Italy), littermate wild type ($Trpa1^{+/+}$) and TRPA1-deficient ($Trpa1^{-/-}$) mice (25–30 g, 5–8 weeks), generated by heterozygotes on a C57BL/6 background

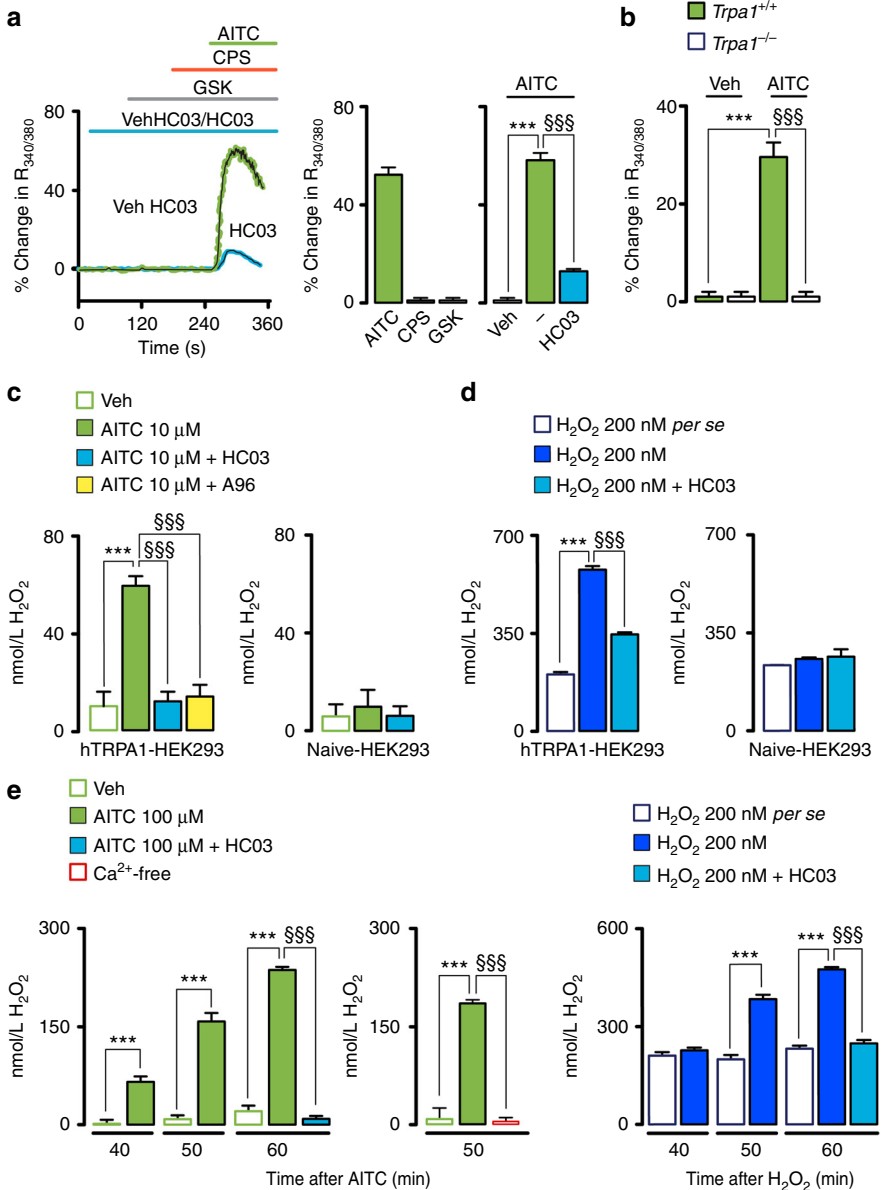

**Fig. 4** Schwann cells expressing TRPA1 release $H_2O_2$. **a** $Ca^{2+}$ responses to AITC (1 mM) in cultured Schwann cells with HC-030031 (HC03, 30μM) or its vehicle (veh, 0.3% DMSO) and to TRPV1- (capsaicin, CPS, 0.5 μM) or TRPV4- (GSK1016790A, GSK, 50 nM) agonists ($n = 25$ cells from 3 independent experiments, ***$P < 0.001$ AITC vs. veh; $^{§§§}P < 0.001$ HC03 vs. AITC; one-way ANOVA followed by Bonferroni post hoc analyses). **b** AITC (1 mM)-evoked calcium response in Schwann cells from $Trpa1^{+/+}$, but not from $Trpa1^{-/-}$ mice ($n = 25$ cells from 3 independent experiments, ***$P < 0.001$ $Trpa1^{+/+}$ AITC vs. $Trpa1^{+/+}$ veh; $^{§§§}P < 0.001$ $Trpa1^{-/-}$ AITC vs. $Trpa1^{+/+}$ AITC; one-way ANOVA followed by Bonferroni post hoc analyses). **c, d** $H_2O_2$ release from hTRPA1-HEK293 or untransfected (naïve-HEK293) cells induced by AITC (10 μM) or $H_2O_2$ (200 nM) and effect of HC03 (30 μM), A-967079 (A96, 30 μM) or respective vehicles (veh, 0.3% DMSO) ($n = 8$ replicates from three independent experiments, ***$P < 0.001$ AITC, $H_2O_2$ vs. veh; $^{§§§}$ $P < 0.001$ AITC, $H_2O_2$ + HC03/A96 vs. AITC, $H_2O_2$; one-way ANOVA followed by Bonferroni post hoc analyses; $H_2O_2$ 200 nM *per se* represents the value of $H_2O_2$ over the time, not in presence of cells). **e** $H_2O_2$ release from cultured mouse Schwann cells evoked by AITC (100 μM) or $H_2O_2$ (200 nM) and effect of HC03 (30 μM) and $Ca^{2+}$-free medium ($Ca^{2+}$-free) ($n = 8$ replicates from three independent experiments, ***$P < 0.001$, veh-AITC/$H_2O_2$ vs. AITC/$H_2O_2$; $^{§§§}P < 0.001$ HC03 vs. AITC, $H_2O_2$; one-way ANOVA followed by Bonferroni post hoc analyses; $H_2O_2$ 200 nM per se represents the value of $H_2O_2$ without cells). Data are represented as mean ± s.e.m

(B6.129P-$Trpa1^{tm1Kykw/J}$; Jackson Laboratories, Bar Harbor, ME, USA)[65], wild type ($Trpv4^{+/+}$) and TRPV4-deficient ($Trpv4^{-/-}$) mice (25–30 g, 5–8 weeks), generated by heterozygotes on a C57BL/6 background[66] and TRPV1-deficient mice ($Trpv1^{-/-}$; B6.129 × 1-$Trpv1^{tm1Jul/J}$) backcrossed with C57BL/6 mice ($Trpv1^{+/+}$) for at least 10 generations (Jackson Laboratories, Bar Harbor, ME, USA; 25–30 g, 5–8 weeks), were used. B6.Cg-Tg(Plp1-CreERT)3Pop/J mice ($Plp1$-$Cre^{ERT}$, Stock No: 005975), expressing a tamoxifen-inducible Cre in myelinating cells (Plp1, proteolipid protein myelin 1)[67], and 129S-Trpa1$^{tm2Kykw/J}$ mice (*floxed TRPA1*, $Trpa1^{fl/fl}$, Stock No: 008650), which possess loxP sites on either side of the S5/S6 transmembrane domains of the $Trpa1$ gene, were obtained from Jackson Laboratories (Bar Harbor, ME, USA). To generate mice in which the $Trpa1$ gene was

conditionally silenced in Schwann cells/oligodendrocytes homozygous $Trpa1^{fl/fl}$ mice were crossed with hemizygous $Plp1$-$Cre^{ERT}$ mice.

The progeny was genotyped by standard PCR for $Cre^{ERT}$ and $Trpa1$ alleles (PCR protocol ID 005975; PCR protocol ID 008650, Jackson Laboratories, Bar Harbor, ME, USA). Both positive or negative mice to $Cre^{ERT}$ and homozygous for floxed $Trpa1$ ($Plp1$-$Cre^{ERT+};Trpa1^{fl/fl}$ and $Plp1$-$Cre^{ERT−};Trpa1^{fl/fl}$, respectively) were treated with intraperitoneal (i.p.) 4-hydroxytamoxifen (1 mg 100 μl$^{-1}$ in corn oil, once a day, for 5 consecutive days)[67] resulting in Cre-mediated ablation of $Trpa1$ in PLP-expressing Schwann cells/oligodendrocytes. Successful Cre-driven deletion of TRPA1 mRNA was confirmed by RT-qPCR. Mice negative to $CreERT$ ($Plp1$-$Cre^{ERT−};Trpa1^{fl/fl}$) were used as control. Mice were allocated to pSNL/sham surgery

or underwent to functional experiments (AITC challenge) 15 days after the last tamoxifen injection. Tissues or cells used to assess the level of TRPA1 expression were collected also 15 days after the last tamoxifen injection.

Animals were housed in a temperature- and humidity-controlled vivarium (12 h dark/light cycle, free access to food and water, 10 animals per cage). Behavioral experiments were performed, after 1 h of animal acclimation, in a quiet,

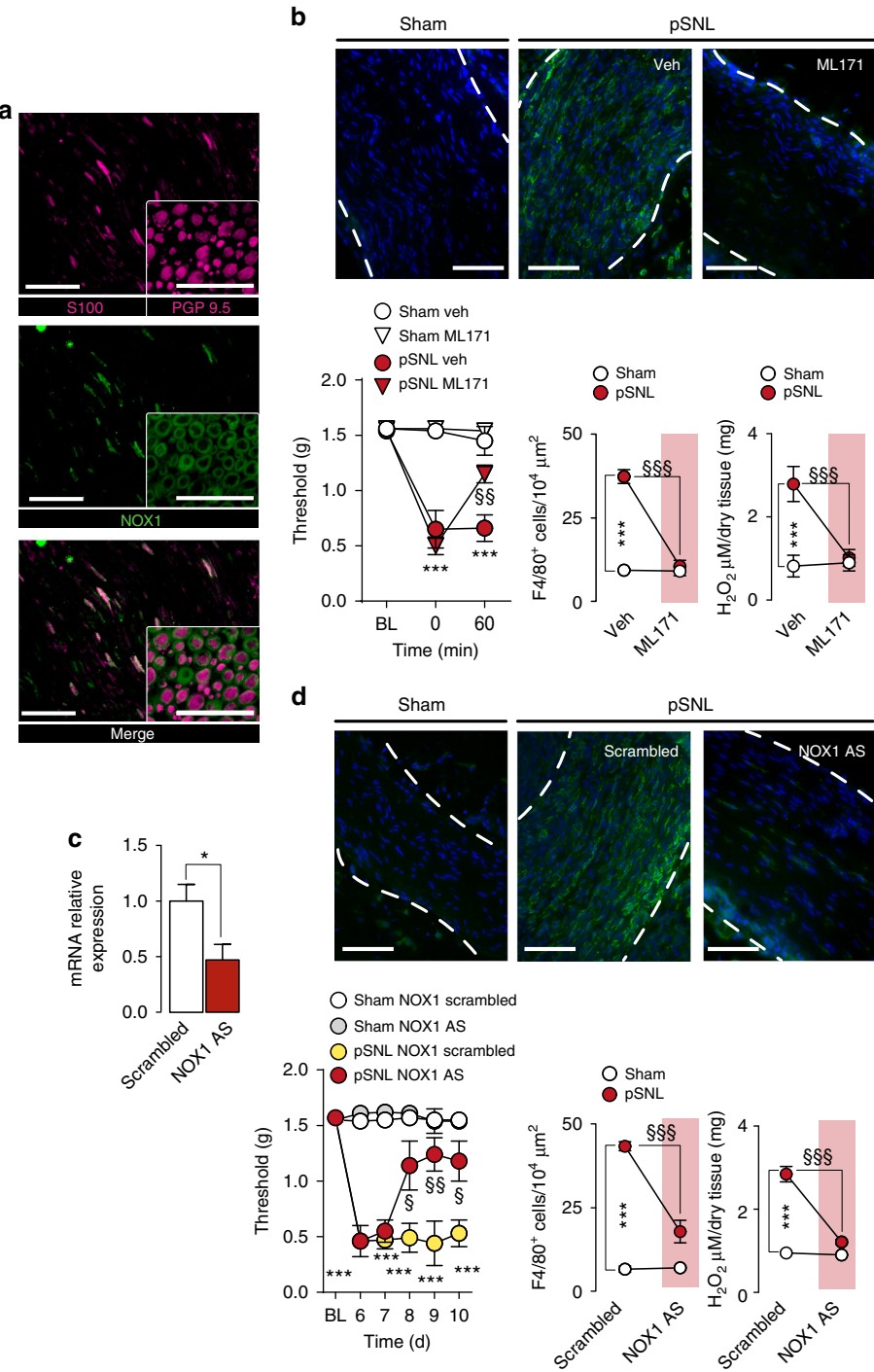

**Fig. 5** NOX1 blockade inhibited pSNL-evoked allodynia and neuroinflammation. **a** Images of S100/PGP9.5 and NOX1 immunofluorescence in sciatic nerves (Scale bars: 50 μm and inset 20 μm). **b** Mechanical allodynia, representative images and F4/80$^+$ cell number, and H$_2$O$_2$ content (at day 10 after surgery) in sham/pSNL mice after NOX1-inhibitor (ML171, 60 mg kg$^{-1}$, i.p) or vehicle (veh, 4% DMSO and 4% tween 80 in isotonic saline) ($n = 6$, ***$P < 0.001$ pSNL-veh vs. sham-veh; $^{§§}P < 0.01$ and $^{§§§}P < 0.001$ pSNL-ML171 vs. pSNL-veh; two-way ANOVA followed by Bonferroni post hoc analyses) (Scale bars: 50 μm; dashed lines, perineurium). **c** NOX1 mRNA relative expression in sciatic nerve after perineural NOX1 antisense oligonucleotides (AS-ODN) or scrambled-ODN (both, 10 nmol 10 μl$^{-1}$) ($n = 3$ replicates from two independent experiments, *$P < 0.05$ AS-ODN vs. scrambled-ODN; unpaired two-tailed Student's t-test). **d** Mechanical allodynia, representative images and F4/80$^+$ cell number, and H$_2$O$_2$ content (at day 10 after surgery) in sham/pSNL mice after NOX1 AS-ODN or scrambled-ODN (both, 10 nmol 10 μl$^{-1}$) ($n = 6$, ***$P < 0.001$ pSNL-NOX1-scrambled vs. sham-NOX1scrambled; $^{§}P < 0.05$, $^{§§}P < 0.01$ and $^{§§§}P < 0.001$ pSNL-NOX1 AS-ODN vs. pSNL-NOX1scrambled; two-way ANOVA followed by Bonferroni post hoc analyses) (Scale bars: 50 μm; dashed lines, perineurium). Data are represented as mean ± s.e.m

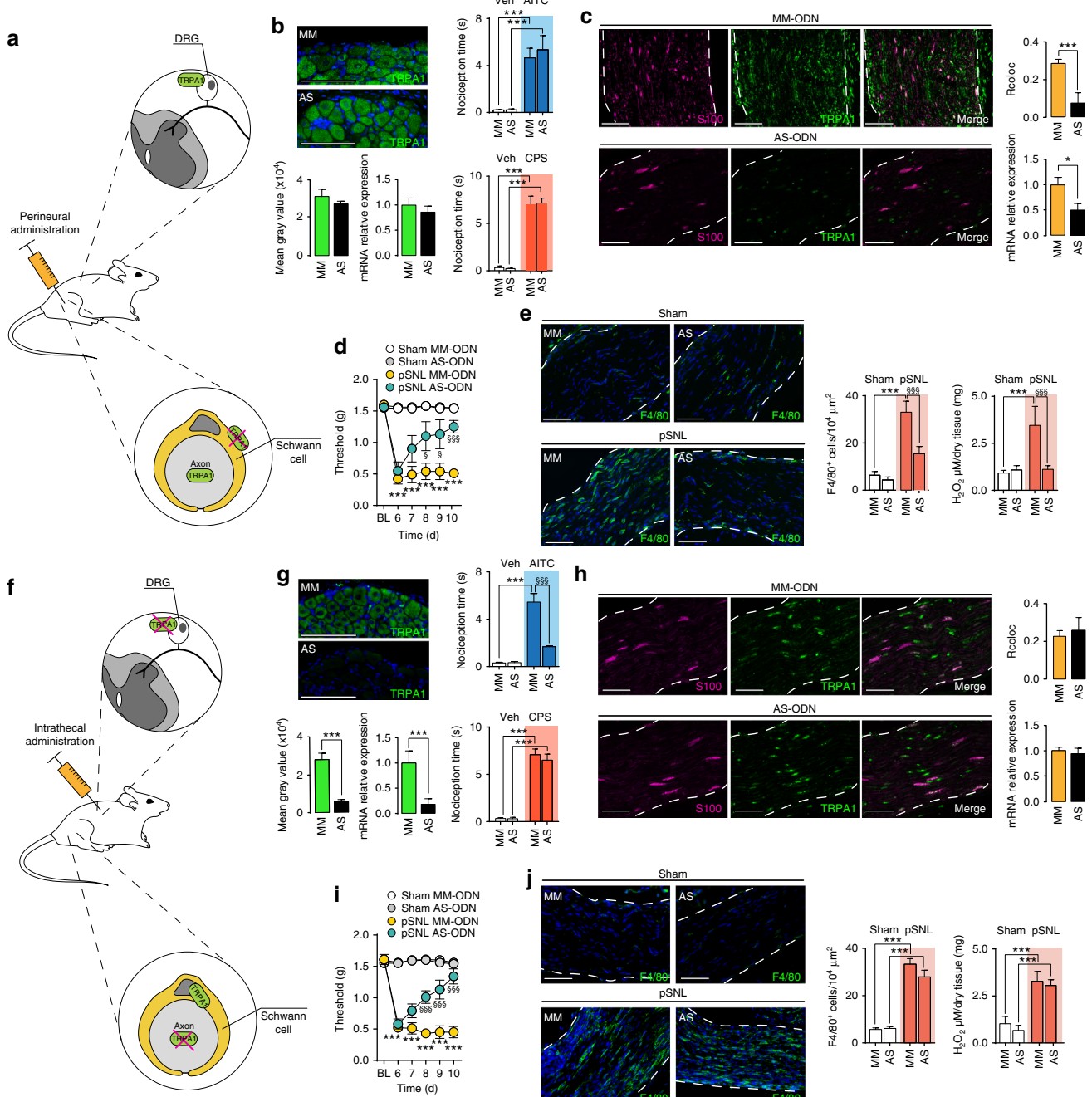

**Fig. 6** Oxidative stress from Schwann cell TRPA1 recruits macrophages and signal pain in C57BL/6 mice. **a**, **f** Schematic representation of perineural/intrathecal injection of TRPA1 antisense/mismatch oligonucleotides (AS/MM-ODN). **b**, **g** TRPA1 immunofluorescence (mean gray value) and TRPA1 mRNA relative expression in DRGs and acute nociception after perineural AITC (20 nmol 10 μl$^{-1}$) or capsaicin (CPS, 1 nmol 10 μl$^{-1}$) following perineural (10 nmol 10 μl$^{-1}$) (**b**) or intrathecal (5 nmol 5 μl$^{-1}$) (**g**) TRPA1 AS/MM-ODN treatment (once/day for 4 consecutive days) in C57BL/6 ($n = 6$, ***$P < 0.001$ MM/AS AITC, CPS vs. MM/AS veh; $^{§§§}P < 0.001$ AS AITC vs. MM AITC; one-way ANOVA followed by Bonferroni post hoc analyses, Scale bars: 20 μm). **c**, **h** Representative images (Scale bars: 50 μm; dashed lines, perineurium), (**j**) colocalization value (Rcoloc) of S100/TRPA1 and mRNA-TRPA1 expression in sciatic nerve after perineural (**c**) and intrathecal (**h**) AS/MM-ODN ($n = 6$, *$P < 0.05$; ***$P < 0.001$ AS vs MM; unpaired two-tailed Student's t-test). **d**, **i** Mechanical allodynia, and (**e**, **j**) representative images, F4/80$^+$-cells, and H$_2$O$_2$-content (at day 10 after surgery) in sham/pSNL mice after perineural (**d**, **e**) and intrathecal (**i**, **j**) AS/MM-ODN ($n = 8$, ***$P < 0.001$ pSNL-MM-ODN vs. sham-MM-ODN; $^{§}P < 0.05$ and $^{§§§}P < 0.001$ pSNL-AS-ODN vs. pSNL-MM-ODN; (**d**, **i**) two-way ANOVA followed by Bonferroni post hoc analyses and (**e**, **j**) one-way ANOVA followed by Bonferroni post hoc analyses) (Scale bars: 50 μm; dashed lines, perineurium). Data are represented as mean ± s.e.m

temperature-controlled room (20–22 °C) between 9 a.m. and 5 p.m. The sample sizes chosen for animal groups were adequately powered to observe the effects based on both our past experience in similar experimental settings and data published by others. Some animals were excluded because of failure to reach the training criteria or mortality. Exclusions for training were based on scores established before starting experiments and routinely used. Animals were

randomized to vehicle(s) or treatment(s) administration. The assessors (scientists who performed *in vitro* and *in vivo* tests), were blinded to the identity (genetic background or allocation to treatment group) of the animals. Identity of the animals was unmasked to assessors only after data collection. Every effort has been made to minimize the discomfort and pain of the animals in each phase of the study. Animals were euthanized with inhaled CO$_2$ plus 10–50% O$_2$. HC-030031 (2-

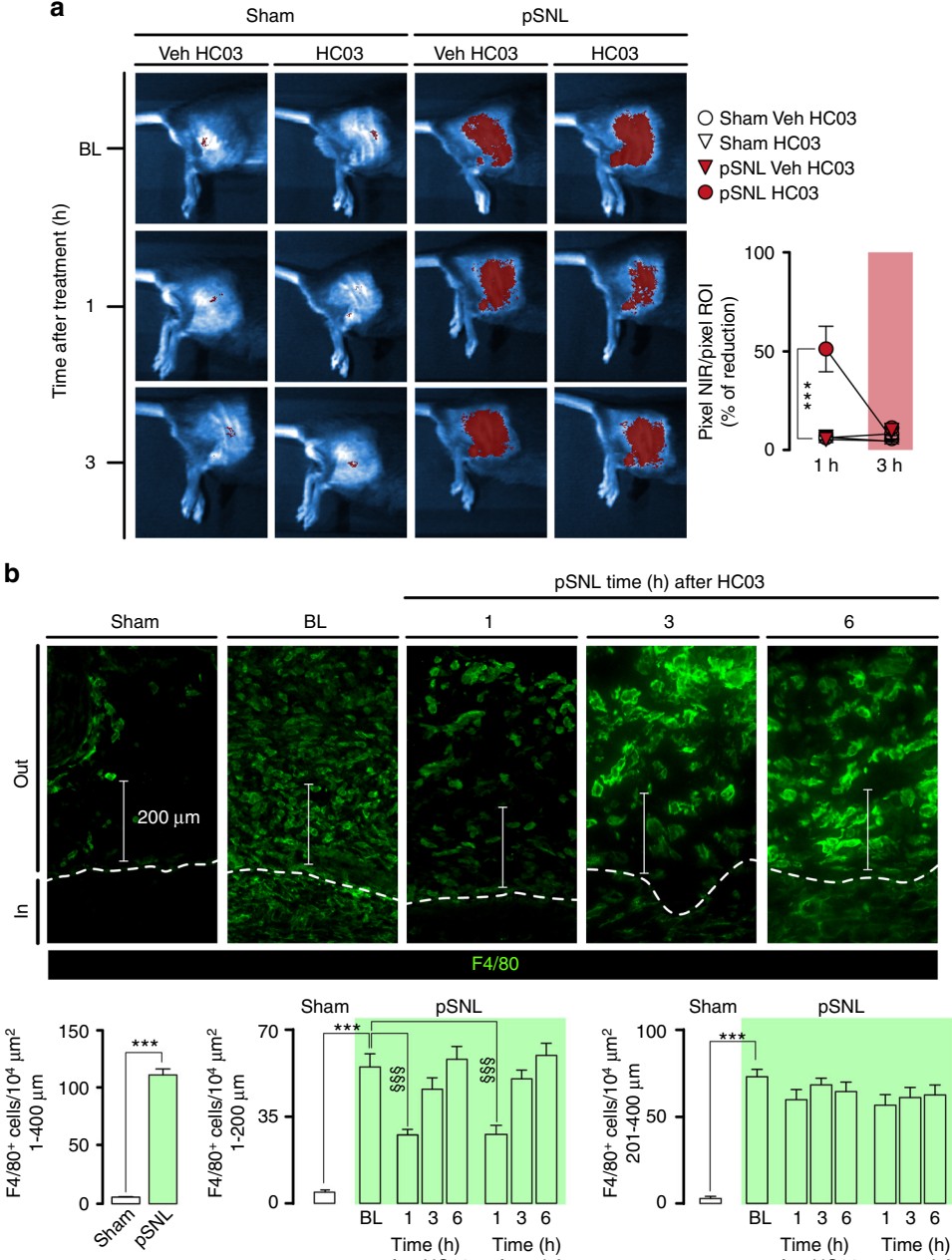

**Fig. 7** TRPA1 blockade and antioxidant reduced the number of fluorescent macrophages accumulated at the site of pSNL. **a** In vivo imaging and quantitative data (NIR area/total ROI) of NIR labeled macrophages (at day 10 after surgery) in sham/pSNL mice at baseline (BL), 1 and 3 h after HC-030031 (HC03, 100 mg kg$^{-1}$, i.p.) ($n = 4$, ***$P < 0.001$ pSNL HC03 vs. pSNL Veh HC03; two-way ANOVA followed by Bonferroni post hoc analyses). **b** Representative images and F4/80$^+$-cell number surrounding the injured nerve trunk (at day 10 after surgery) of sham/pSNL-mice at BL and 1, 3 and 6 h after HC03 or alpha-lipoic acid (αLA) (both, 100 mg kg$^{-1}$, i.p.). ($n = 4$, ***$P < 0.001$ pSNL vs sham; $^{§§§}P < 0.001$ pSNL HC03, αLA vs. pSNL veh; one-way ANOVA followed by Bonferroni post hoc analyses) (Scale bars, 200 μm; inside (in) and outside (out) sciatic nerves). Data are represented as mean ± s.e.m

(1,3-Dimethyl-2,6-dioxo-1,2,3,6-tetrahydro-7H-purin-7-yl)-N-(4-isopropylphenyl) acetamide) was synthesized as previously described[68]. If not otherwise indicated, reagents were obtained from Sigma-Aldrich (Milan, Italy).

**Partial ligation of the sciatic nerve.** Partial ligation of the sciatic nerve (pSNL) was performed in C57BL/6, in Trpa1$^{+/+}$ or Trpa1$^{-/-}$, in Trpv1$^{+/+}$ or Trpv1$^{-/-}$, in Trpv4$^{+/+}$ or Trpv4$^{-/-}$ and in Plp1-Cre$^{ERT}$;Trpa1$^{fl/fl}$ or control mice as previously described[69]. Briefly, mice were anesthetized with i.p. injection of a mixture of ketamine (90 mg kg$^{-1}$) and xylazine (3 mg kg$^{-1}$), the right sciatic nerve was exposed, and a partial ligation was made by tying one-third to one half of the dorsal portion of the sciatic nerve. In sham-operated mice, used as controls, the right sciatic nerve was exposed, but not ligated. Mice were monitored, adequately rehydrated, and maintained in a controlled temperature (37 °C) until fully recovered from anesthesia.

**Experimental design.** C57BL/6, Trpa1$^{+/+}$ or Trpa1$^{-/-}$, Trpv1$^{+/+}$ or Trpv1$^{-/-}$, Trpv4$^{+/+}$, or Trpv4$^{-/-}$ and Plp1-Cre$^{ERT}$;Trpa1$^{fl/fl}$ or control mice were randomly allocated to pSNL or sham surgery. Ten days after surgery, C57BL/6 mice were randomly allocated to the treatment group with i.p. administration of TRPA1 selective antagonists, HC-030031 (HC03) or A-967079 [(1E,3E)-1-(4-Fluorophenyl)-2-methyl-1-penten-3-one oxime] (A96) and antioxidants, α-lipoic acid (αLA) or phenyl-N-tert-butylnitrone (PBN) (all 100 mg kg$^{-1}$), the nicotinamide adenine dinucleotide phosphate (NADPH) oxidase (NOX) inhibitors, GKT137831 [2-(2-chlorophenyl)-4-[3-(dimethylamino)phenyl]-5-methyl-1H-pyrazolo[4,3-c]pyridine-3,6(2H,5H)-dione] (GKT, 60 mg kg$^{-1}$, NOX1/4 selective), 2-acetylphenothiazine (ML171, 60 mg kg$^{-1}$, NOX1 selective), gp91ds-tat peptide (10 mg kg$^{-1}$, NOX2 selective) or vehicles (all 4% dimethyl sulfoxide, DMSO, 4% tween 80 in isotonic saline, NaCl 0.9%; PBN was dissolved in NaCl 0.9%).

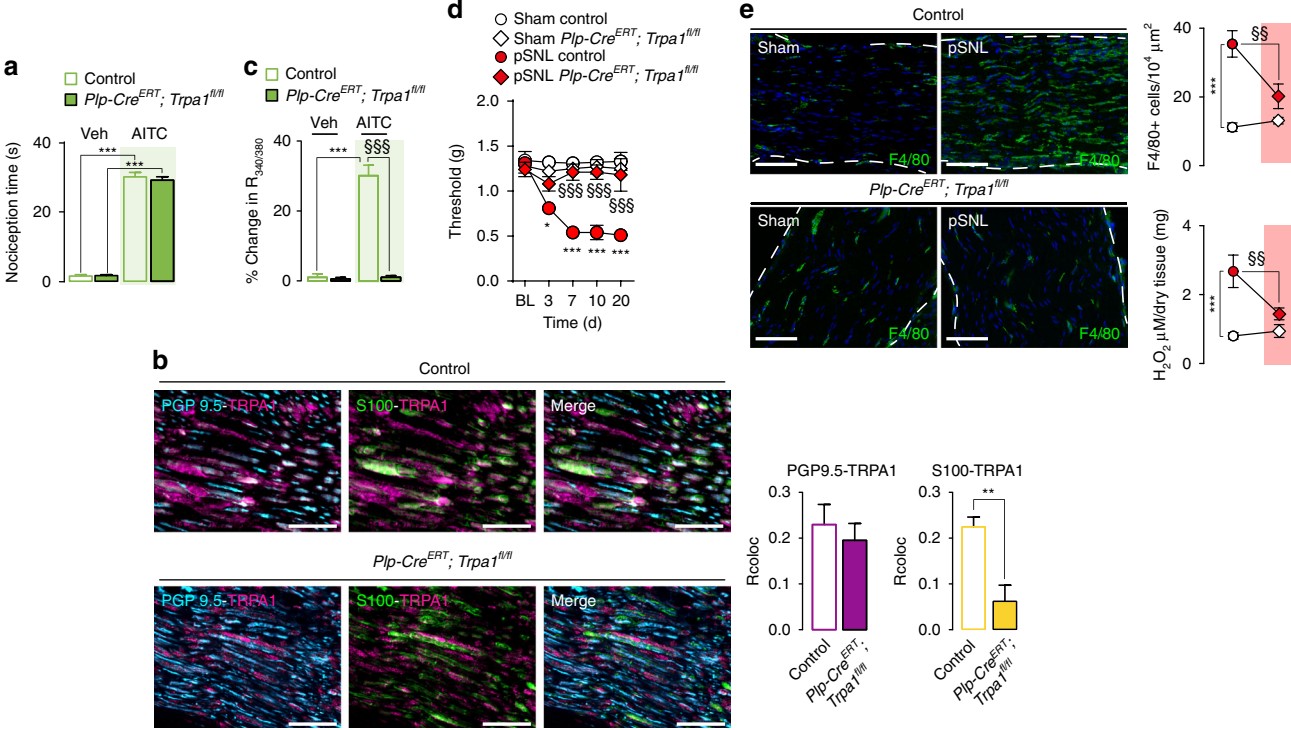

**Fig. 8** Plp1-Cre/ERT–mediated *Trpa1* deletion from Schwann cells prevented the partial sciatic nerve ligation (pSNL)-evoked allodynia and neuroinflammation. **a** Acute nociception response induced by intraplantar AITC (10 nmol 20 µl⁻¹) or vehicle (veh, 0.5% DMSO) in *Plp1-Cre^ERT;Trpa1^fl/fl* and control mice (*n* = 6, \*\*\**P* < 0.001 AITC vs. veh; one-way ANOVA followed by Bonferroni post hoc analyses). **b** Triple immunofluorescence staining and colocalization value (Rcoloc) of TRPA1, PGP9.5, and S100 (Scale bars: 20 µm) in sciatic nerve trunks obtained from *Plp1-Cre^ERT;Trpa1^fl/fl* (*n* = 5) and control mice (*n* = 4), (\*\**P* < 0.01; *Plp1-Cre^ERT;Trpa1^fl/fl* vs. control; unpaired two-tailed Student's *t*-test). **c** Ca²⁺ responses to AITC (1 mM) or vehicle (veh, 1% DMSO) in cultured Schwann cells from sciatic nerve trunks of *Plp1-Cre^ERT;Trpa1^fl/fl* and control mice (*n* = 25 cells/two independent experiments, \*\*\**P* < 0.001 AITC vs. veh; §§§*P* < 0.001 AITC *Plp1-Cre^ERT;Trpa1^fl/fl* vs. AITC control; one-way ANOVA followed by Bonferroni post hoc analyses). **d** Mechanical allodynia, and (**e**) representative images, F4/80⁺ cells, and H₂O₂ content (at day 10 after surgery) in sham/pSNL *Plp1-Cre^ERT;Trpa1^fl/fl* and control mice (*n* = 8, \**P* < 0.05 and \*\*\**P* < 0.001 pSNL control vs. sham control; §§*P* < 0.01 and §§§*P* < 0.001 pSNL *Plp1-Cre^ERT;Trpa1^fl/fl* vs. pSNL control; (**d**) two-way ANOVA followed by Bonferroni post hoc analyses and (**e**) one-way ANOVA followed by Bonferroni post hoc analyses) (Scale bars: 50 µm). Data are represented as mean ± s.e.m

In different experiments, at days 6, 7, 8 and 9 after surgery, pSNL or sham C57BL/6 mice were randomly allocated to the groups receiving perineural (p.n.) treatment (10 nmol 10 µl⁻¹) of TRPA1, NOX1, NOX2 and NOX4 antisense (AS), mismatch (MM) oligonucleotide (ODN) or scrambled control. TRPA1 AS-ODN sequence was 5′-TATCGCTCCACATTGCTAC-3′, TRPA1 MM-ODN sequence was 5′-ATTCGCCTCACATTGTCAC-3′[44], NOX1 AS-ODN sequence was 5′-TTA ACCAGCCAGTTTCCCATTG-3′, NOX2 AS-ODN sequence was 5′-TTCACAG CCCAGTTCCCCATG T-3′, NOX4 AS-ODN sequence was 5′-TTGGCCAGCCAG CTCCTCCA-3′, NOX1 and NOX2 scrambled p47phox sequence control was 5′-T GAGGCTCC GTCCGCTGGAGCG-3′, NOX4 scrambled sequence was 5′-CGTCA CGCTCAGCTCACCGT-3′[70,71]. Perineural injections were performed as previously reported[11,72]. Briefly, compounds were injected into the region surrounding the sciatic nerve at high thigh level of right hind limbs without skin incision in a volume of 10 µl using a microsyringe fitted with a 30-gauge needle. Other mice were treated with intratechal (i.t.) treatment (5 nmol 5 µl⁻¹) of TRPA1 AS/MM-ODN[44]. Phosphorothioate-modified ODNs, which is reported to be more effective in the rate of uptake and more specific in binding the target sequence, generally increasing ODNs potency[73] were used for both perineural and intrathecal treatments. Some mice treated with TRPA1 AS/MM-ODN (p.n. or i.t.) were tested for the acute nociceptive response to AITC (20 nmol 10 µl⁻¹, p.n.) or CPS (1 nmol 10 µl⁻¹, p.n.).

pSNL/sham C57BL/6 mice were treated (i.p.) with an antibody directed to the chemokine (C–C motif) ligand 2 (CCL2) (CCL2-Ab) (R&D system, Minneapolis, USA) with two different protocols. In the first, the CCL2-Ab or vehicle (IgG2B Isotype Control, R&D system, Minneapolis, USA) were given at day 7, 8 and 9 (40 µg 200 µl⁻¹, i.p.) after surgery. In the second, one single triple-dose (120 µg 200 µl⁻¹, i.p.) or vehicle were given at day 10 after surgery. To transiently deplete the monocyte/macrophage population, pSNL or sham C57BL/6 mice received liposome-encapsulated clodronate (LCL, 5 mg ml⁻¹ i.p., ClodronateLiposomes.com Amsterdam, The Netherlands) or vehicle (liposome-encapsulated phosphate buffer saline) at day 7, 8 and 9 after surgery.

C57BL/6 mice received subcutaneous (s.c.) resiniferatoxin (RTX, 50 µg kg⁻¹) or its vehicle (10% ethanol and 10% tween 80 in isotonic saline)[55]. pSNL or sham surgery was carried out five days after RTX treatment when the eye wiping test responses to both capsaicin (CPS, 1 nmol 5 µl⁻¹) and AITC (20 nmol 5 µl⁻¹) were abolished (Supplementary Fig. 4A). In different experiments, perineural (p.n.) CCL2 was given to *Trpa1^+/+* or *Trpa1^−/−* mice and C57BL/6 mice (0.1, 0.5 and 1 µg 10 µl⁻¹), which after 2 h received i.p. HC03 or αLA (both 100 mg kg⁻¹) or vehicles. Some C57BL/6 mice were treated with LCL (5 mg ml⁻¹ i.p.) or vehicle for 3 consecutive days and then CCL2 (1 µg 10 µl⁻¹, p.n.) was administered.

**Assessment of pain-like behaviors.** Mechanical allodynia, cold hypersensitivity and thermal hyperalgesia were assessed before (baseline) and 3, 7, 10, 20 days after pSNL or sham surgery in C57BL/6, *Trpa1^+/+*, *Trpa1^−/−*, *Trpv1^+/+*, *Trpv1^−/−*, *Trpv4^+/+*, *Trpv4^−/−*, *Plp1-Cre^ERT;Trpa1^fl/fl*, and control mice. In C57BL/6 mice, mechanical allodynia was assessed before and after the various pharmacological interventions at different time points. To measure nociceptive responses, mice were placed in a Plexiglas chamber immediately after p.n. injection of AITC or CPS, or intraplantar injection of AITC (10 nmol 20 µl⁻¹) and the total time spent licking and lifting (nociception time, s) the injected right hind limb and paw was recorded for 5 min. Eye wiping was assayed after ocular instillation of CPS or AITC or vehicles (2 and 4% DMSO, respectively)[47]. Mechanical threshold was measured by the up-and-down paradigm[74,75]. Cold hypersensitivity was assessed by measuring the acute nociceptive response to acetone-evoked evaporative cooling[24].

Thermal hyperalgesia was measured by exposing the mid plantar surface of the hind paw to a beam of radiant heat through a transparent surface, using a plantar analgesimeter for paw stimulation (Ugo Basile, Comerio, Italy). Paw withdrawal latency was recorded as the time from onset of the thermal stimulus to the withdrawal response. In each paw mean withdrawal latency of three measures was calculated. The interval between trials on the same paw was at least 5 min. The

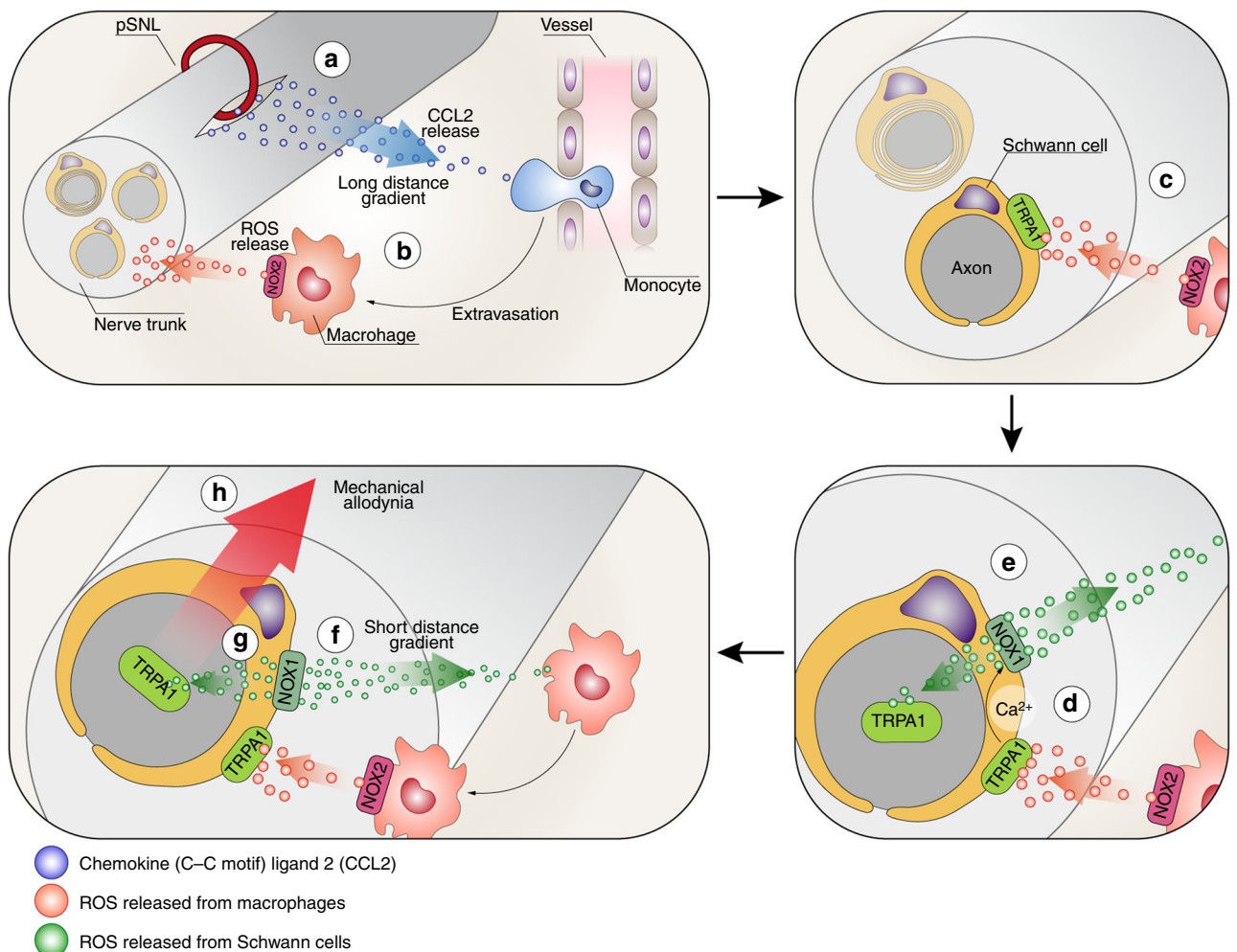

**Fig. 9** Cellular and molecular events contributing to TRPA1-mediated mechanical allodynia and neuroinflammation in a neuropathic pain model. Partial sciatic nerve ligation (pSNL) by releasing CCL2 (**a**) promotes the extravasation of hematogenous monocytes (**b**) that, via their rapid NOX2-dependent oxidative burst (red dots) target the TRPA1 channel localized in Schwann cells (**c**). TRPA1 activation in Schwann cells evokes a $Ca^{2+}$-dependent, NOX1-mediated (**d**) prolonged $H_2O_2$ (green dots) generation (**e**) with a dual function. The outward $H_2O_2$ release (**f**) produces a space-scaled gradient that determines the final macrophage influx to the injured nerve trunk, whereas the inward $H_2O_2$ release (**g**) targets nociceptor TRPA1 to produce mechanical allodynia (**h**). ROS reactive oxygen species

cutoff latency was set at 20 s to avoid tissue damage in case of failure to remove the paw.

**CCL2 enzyme-linked immunosorbent assay.** The CCL2 content in the sciatic nerve (ipsilateral to the surgery) was measured in $Trpa1^{+/+}$, $Trpa1^{-/-}$ and C57BL/6 mice 10 days after pSNL/sham surgery and 60 min after treatment (C57BL/6 mice) with HC03, αLA, or their vehicles and CCL2-Ab or IgG2B control (single and triple administration) or LCL, by using a mouse CCL2/MCP-1 quantikine ELISA Kit (R&D system, Minneapolis, USA). Samples were homogenized at 4 °C in PBS containing a protease inhibitor cocktail tablet (Roche Diagnostics, Mannheim, Germany). The homogenate was then centrifuged ($10,000 \times g$, 20 min, 4 °C); supernatants were collected and assayed according to the manufacturer's instructions. The concentration of CCL2 was expressed in pg/mg of total protein content[76].

**Measurement of $H_2O_2$ content in tissue.** The $H_2O_2$ content of the sciatic nerve (ipsilateral to the surgery) was firstly determined in C57BL/6 mice at day 3, 7, 10 and 20 after pSNL/sham surgery. Then, all the measurements were performed at day 10 after pSNL/sham surgery and 60 min after treatment with HC03, αLA, GKT, PBN, ML171, gp91ds-tat peptide, or anti-CCL2 antibody. In $Trpa1^{+/+}$, $Trpa1^{-/-}$, $Plp1-Cre^{ERT};Trpa1^{fl/fl}$, control, and C57BL/6 mice pretreated with RTX or treated with TRPA1, NOX1, NOX2, NOX4 AS/MM-ODN, anti-CCL2 antibody or LCL, $H_2O_2$ content was assessed 10 days after pSNL/sham surgery. $H_2O_2$ was determined by using the Amplex Red® assay (Invitrogen, Milan, Italy). Briefly, sciatic nerves were rapidly removed and placed into modified Krebs/HEPES buffer (composition in mmol l$^{-1}$: 99.01 NaCl, 4.69 KCl, 2.50 CaCl$_2$, 1.20 MgSO$_4$, 1.03

KH$_2$PO$_4$, 25.0 NaHCO$_3$, 20.0 Na-HEPES, and 5.6 glucose [pH 7.4]). Samples were minced and incubated with Amplex red (100 μM) and HRP (1 U ml$^{-1}$) (60 min, 37 °C) in modified Krebs/HEPES buffer protected from light[77]. Fluorescence excitation and emission were at 540 and 590 nm, respectively. $H_2O_2$ production was calculated using $H_2O_2$ standard and expressed as μmol l$^{-1}$ of mg of dry tissue.

**Cell culture.** HEK293 cells stably transfected with the cDNA for human TRPA1 (hTRPA1-HEK293, kindly donated by A.H. Morice, University of Hull, Hull, UK) and naive untransfected HEK293 cells (American Type Culture Collection, Manassas, VA, USA; ATCC® CRL-1573™), were cultured as previously described[78]. For all cell lines, the cells were used when received without further authentication. Schwann cells were isolated from sciatic nerves of C57BL/6, $Trpa1^{+/+}$, $Trpa1^{-/-}$, $Plp1-Cre^{ERT};Trpa1^{fl/fl}$, or control mice. Briefly, the epineurium was removed, and nerve explants were divided into 1 mm segments and dissociated enzymatically using collagenase (0.05%) and hyaluronidase (0.1%) in HBSS (2 h, 37 °C). Cells were collected by centrifugation (800×rpm, 10 min, RT) and the pellet was resuspended and cultured in DMEM containing: 10% FCS, 2 mM L-glutamine, 100 U ml$^{-1}$ penicillin/100 mg ml$^{-1}$ streptomycin or 50 mg ml$^{-1}$ gentamycin. Three days later, cytosine arabinoside (10 mM) was added to remove fibroblasts. To enhance Schwann cell proliferation, forskolin (2 μM) was added to the culturing medium[79].

To obtain cultured peritoneal macrophages, C57BL/6 mice were i.p. injected with thioglycolate (3%, 1 ml). After 3 days, cells were harvested from sacrificed animals by peritoneal lavage for a total of 10 ml PBS and centrifuged ($400 \times g$, 10 min, 4 °C). Cells were cultured in DMEM supplemented with 10% FBS. After incubation at 37 °C for 24 h, non-adherent cells were removed by repeated

washing[80]. Before each experiment, cells were tested with specific kits for cells mycoplasma contamination based on PCR (EMK090020, N-GARDE kit, Euroclone, Milan, Italy).

**Measurement of $H_2O_2$ released from cells**. $H_2O_2$ was determined by using the Amplex Red assay (Invitrogen, Milan, Italy). hTRPA1-HEK293 or naive untransfected HEK293, Schwann cells or peritoneal macrophages were plated in 96-well clear bottom black ($5 \times 10^5$ cells well$^{-1}$) and maintained in 5% $CO_2$ and 95% $O_2$ (24 h, 37 °C). The cultured medium was replaced with Krebs-Ringer phosphate (KRP, composition in mmol l$^{-1}$: 2 $CaCl_2$; 5.4 KCl; 0.4 $MgSO_4$; 135 NaCl; 10 D-glucose; 10 HEPES [pH 7.4]) added with HC03, A96 (both 30 µM) or vehicle (0.3% DMSO) for 10 min at RT. Peritoneal macrophages were incubated with GKT (100 nM) or gp91ds-tat (0.1–100 nM) for 20 min. hTRPA1-HEK293, naive HEK293 or Schwann cells were stimulated with AITC (10 and 100 µM, respectively), $H_2O_2$ (200 nM) or their vehicle (0.01% DMSO or KRP, respectively), peritoneal macrophages were stimulated with phorbol myristate acetate (PMA, 20 nM) or vehicle (0.00001% DMSO diluted in KRP) added with Amplex red (50 µM) and HRP (1 U ml$^{-1}$), and maintained for 30 min at RT protected from light. Some experiments in Schwann cells were performed in $Ca^{2+}$-free KRP containing EDTA (1 mM). Signal was detected 60 min (hTRPA1-HEK293/naïve-HEK293) or 40, 50, and 60 min (Schwann cells) after exposure to the stimulus. $H_2O_2$ release was calculated using $H_2O_2$ standards and expressed as nmol l$^{-1}$.

**Calcium imaging**. Schwann cells and macrophages were plated on glass coated (poly-L-lysine, 8.3 µM) coverslips and intracellular calcium response was measured as previously reported[81]. Schwann cells were challenged with the selective TRPA1 agonist, AITC (1 mM), and the selective TRPV1 and TRPV4 agonists, CPS (0.5 µM) and GSK1016790A (GSK, 50 nM), respectively. Results are expressed as % increase in Ratio$_{340/380}$ over baseline normalized to the maximum effect induced by ionomycin (5 µM) added at the end of each experiment (% change in R$_{340/380}$). Macrophages were stimulated with fresh medium containing 100 ng ml$^{-1}$ LPS, then incubated at 37 °C for 6, 12, 18, 24, 36 and 48 h, before being challenged with AITC (1 mM) and ionomycin (5 µM). Results are expressed as Ratio$_{340/380}$.

**Immunofluorescence and confocal microscopy**. Anesthetized mice were transcardially perfused with PBS, followed by 4% paraformaldehyde. The sciatic nerves (ipsilateral to the surgery) or dorsal root ganglia (DRGs, L4-L6) were removed, postfixed for 24 h, and paraffin embedded or cryoprotected overnight at 4 °C in 30% sucrose until cryosectioning. Cryosections (10 µm) were stained with hematoxylin and eosin (H&E) for histological examination or incubated with the following primary antibodies: F4/80 (MA516624, rat monoclonal (Cl:A3-1), 1:50, Thermo Fisher Scientific, Rockford, USA), CD8 (ab22378, rat monoclonal (YTS169.4), 1:200, Abcam, Cambridge, UK) and Ly6G (ab25377, rat monoclonal (RB6-8C5), 1:200, Abcam, Cambridge, UK) (1 h, RT), diluted in fresh blocking solution (PBS, pH 7.4, 10% normal goat serum, NGS). Formalin fixed paraffin-embedded sections (5 µm) were incubated with the following primary antibodies: protein gene product 9.5 (PGP9.5, ab8189, mouse monoclonal [13C4/I3C4], 1:600, Abcam, Cambridge, UK), TRPA1 (ab58844, rabbit polyclonal, 1:400, Abcam, Cambridge, UK), S100 (ab14849, mouse monoclonal (4B3), 1:300, Abcam, Cambridge, UK), SOX10 (ab216020, mouse monoclonal (SOX10/1074), 1:300, Abcam, Cambridge, UK), 4- HNE (ab48506, mouse monoclonal (HNEJ-2), 1:40, Abcam, Cambridge, UK) or NOX1 (ab131088, rabbit polyclonal, 1:250, Abcam, Cambridge, UK) (1 h, RT) diluted in antibody diluent (Roche Diagnostics, Mannheim, Germany). Sections were then incubated with fluorescent secondary antibodies: polyclonal Alexa Fluor 488, polyclonal Alexa Fluor 594, polyclonal Alexa Fluor 546, and polyclonal Alexa Fluor 647 (1:600, Invitrogen, Milan, Italy) (2 h, RT, protected from light). Sections were coverslipped using a water-based mounting medium with 4′6′-diamidino-2-phenylindole (DAPI) (Abcam, Cambridge, UK). The analysis of negative controls (non-immune serum) was simultaneously performed to exclude the presence of non-specific immunofluorescent staining, cross-immunostaining, or fluorescence bleed-through. Tissues were visualized and digital images were captured using an Olympus BX51 or confocal scan a LEICA TCS SP5. High power 3D renderings of the images were obtained using ImageJ 3D viewer.

Direct counting of F4/80+ cells was performed in 10$^4$ µm$^2$ boxes in the sciatic nerve (inside the nerve trunk) in: $Trpa1^{+/+}$, $Trpa1^{-/-}$, Plp1-Cre$^{ERT}$;$Trpa1^{fl/fl}$, and control mice, 10 days after pSNL/sham surgery, and in pSNL/sham C57BL/6 mice at day 10 after surgery following treatment with RTX, CCL2-Ab, LCL and TRPA1, NOX1, NOX2, NOX4 AS/MM-ODN and at different time points after administration of A96, αLA, GKT, PBN, ML171, gp91ds-tat peptide or CCL2-Ab. In some samples, direct counting of F4/80+ cells was performed in pSNL/sham C57BL/6 mice in 10$^4$ µm$^2$ boxes outside the sciatic nerve trunk at two different distances (~0–200 µm and ~ 200–400 µm from the epineurium) before and after HC03 or αLA. Direct counting of CD8+ and Ly6G+ cells was performed in 10$^4$ µm$^2$ boxes in the sciatic nerve (inside the nerve trunk) in pSNL/sham C57BL/6 mice at day 10 after surgery. The counting was performed by an operator blinded to drug treatment and timing.

TRPA1 staining in DRG was evaluated as the fluorescence intensity measured by an image processing software (ImageJ 1.32J, National Institutes of Health, Bethesda, USA). The Pearson correlation (Rcoloc) value for TRPA1 and S100 in the colocalization studies were calculated using the colocalization Plugin of the

ImageJ software[82]. Schwann cells were grown on glass coated (poly-L-lysine, 8.3 µM) coverslips and cultured for 2–3 days before being used for staining. Cells were then fixed in ice-cold methanol/acetone (5 min at −20 °C), washed with PBS and blocked with NGS (10%) (1 h, RT). The cells were then incubated with the primary antibodies (TRPA1, ab58844, rabbit polyclonal, 1:400; S100, ab14849, mouse monoclonal (4B3), 1:300; SOX10, ab216020, mouse monoclonal (SOX10/1074), 1:300, Abcam, Cambridge, UK) (1 h, RT). Cells were then incubated with fluorescent secondary antibodies (1:600, polyclonal Alexa Fluor 488, and polyclonal Alexa Fluor 594, Invitrogen, Milan, Italy) (2 h, RT) and mounted using water-based mounting medium DAPI (Abcam, Cambridge, UK). Cells were visualized and digital images were captured using an Olympus BX51.

**Real-time PCR**. RNA was extracted from cultured Schwann cells or peritoneal macrophages obtained from C57BL/6 mice, and from the sciatic nerve or L4-L6 DRGs (ipsilateral to the surgery) of pSNL C57BL/6 mice after TRPA1, NOX1 and NOX4 scrambled/AS/MM-ODN (i.t. or p.n.) To avoid the confounding contribution of NOX2 mRNA from invading macrophages, for this analysis RNA was extracted from the sciatic nerve (ipsilateral to the surgery) of sham C57BL/6 mice. RNA was also extracted from sciatic nerve of Plp1-Cre$^{ERT}$;$Trpa1^{fl/fl}$ and control. The standard Trizol extraction method was used. RNA concentration and purity was assessed spectrophotometrically by measuring the absorbance at 260 and 280 nm. The RNA (100 ng) was reverse-transcribed using the iScript cDNA Synthesis kit (Bio-Rad, Hercules, USA) according to the manufacturer's protocol. For relative quantification of mRNA, real-time PCR was performed on Rotor Gene Q (Qiagen, Hilden, GE). The sets of primers-probes were as follows: 18S-FW (forward): 5′-CGCGGTTCTATTTTGTTGGT-3′, 18S-RE (reverse): 5′-AGTCGG CATCGTTTATGGTC-3′ (NCBI Ref Seq: NR_003278.3); TRPA1-FW: 5′-CAGG ATGCTACGGTTTTTTTCATTACT-3′, TRPA1-RE: 5′-GCATGTGTCAATGTTT GGTACTTCT-3′ (NCBI Ref Seq: NM_177781.4); S100-FW: 5′-TGGATGAAAAC GGAGATGGGG-3′, S100-RE: 5′-ACAGACTGTGCTCAACTGGT-3′ (NCBI Ref Seq: NM_011309); SOX10-FW: 5′-AGATCCAGTTCCGTGTCAATAA-3′, SOX10-RE:5′GCGAGAAGAAGGCTAGGTG-3′ (NCBI Ref Seq: U70441.1); NOX1-FW: 5′-CACTCACCAATGCCCAGGAT-3′, NOX1-RE: 5′-TGGAAGCAA AGGGAGTGACC-3′ (NCBI Ref Seq: NM_172203.2); NOX2-FW: 5′-GAGGTTG GTTCGGTTTTGGC-3′, NOX2-RE: 5′-CAGGAGCAGAGGTCAGTGTG-3′ (NCBI Ref Seq: NM_007807.5); NOX4-FW: 5′-TGTTGGGCCTAGGATTGTGT-3′, NOX4-RE: 5′-TCCTGCTAGGGACCTTCTGT-3′ (NCBI Ref Seq: NM_015760.5); F4/80-FW: 5′-CCCAGCTTATGCCACCTGCA-3′, F4/80-RE: 5′-TCCAGGCCCTGGAACATTGG-3′ (NCBI Ref Seq: NC_000083.6); Floxed TRPA1-FW: 5′-GGGCAGCTTATTGCCTTCAC-3′, Floxed TRPA1-RE: 5′-TTGC GTAAGTACCCAGAGTGGC-3′ (NCBI Ref Seq: NM_177781.4)

The chosen reference gene was the 18S. The SsoAdvanced™ Universal SYBR Green Supermix (Bio-Rad, Hercules, USA) was used for amplification, and the cycling conditions were the following: samples were heated to 95 °C for 1 min followed by 40 cycles of 95 °C for 10 s, and 65 °C for 20 s. PCR reaction was carried out in triplicate. Relative expression of TRPA1 mRNA was calculated using the $2^{-\Delta(\Delta CT)}$ comparative method, with each gene normalized against the internal endogenous reference 18S gene for the same sample.

**Protein extraction and western immunoblot assay**. Cultured Schwann cells and DRGs (L4–L6) neurons were homogenized in a lysis buffer containing (mM): 50 Tris, 150 NaCl, 2 EGTA, 100 NaF, 1 $Na_3VO_4$, 1% Nonidet P40 (pH 7.5) and complete protease inhibitor cocktail (Roche Diagnostics, Mannheim, Germany). Lysates were centrifuged at $14,000 \times g$ at 4 °C for 45 min. Protein concentration in supernatants was determined using DC protein assay (Bio-Rad, Milan, Italy). Samples with equal amount of protein (20 µg) were then separated by NuPAGE 4–12% Bis–Tris gel electrophoresis (Life Technologies, Carlsbad, USA), and the resolved proteins were transferred to a polyvinylidenedifluoride membrane (Merck Millipore Billerica, USA). Membranes were incubated with 5% dry milk in Tris buffer containing 0.1% Tween 20 (TBST; 20 mM Tris at pH 7.5, 150 mM NaCl) for 1 h at RT, and incubated with TRPA1 (NB110-40763, rabbit polyclonal, 1:200, Novus Biologicals, Littleton, USA) or β-actin (mouse monoclonal primary antibody, 1:6000, Thermo Scientific, Rockford, USA) antibodies, at 4 °C overnight. Membranes were then probed with goat anti-mouse or donkey anti-rabbit IgG conjugated with horseradish peroxidase (HRPO) (Bethyl Laboratories Inc., Cambridge, UK) for 1 h at RT. Finally, membranes were washed three times with TBST, and bound antibodies were detected using chemiluminescence reagents (ECL, Pierce, Thermo Scientific, Rockford, USA). The density of specific bands was measured using an image processing program (ImageJ 1.32J, National Institutes of Health, Bethesda, USA) and normalized to β-actin[24]. The uncropped scan of the blot is reported in the Supplementary Fig. 7.

**Live animal imaging**. Macrophage localization in vivo was obtained by NIR imaging of the fluorescent label macrophage mice by using PhotonImager (Biospace Laboratory, Paris, France)[83]. Mouse thioglycollate-elicited peritoneal macrophages were harvested (up to $250 \times 10^6$ cells per ml) and incubated for 15 min at RT with VivoTrack 680 (PerkinElmer, Inc., Waltham, USA), dissolved in sterile PBS, washed, centrifuged ($400 \times g$, 10 min) and diluted to a final concentration of $5 \times 10^6$ cells 40 µl$^{-1}$. Retro-orbital vein injection (40 µl) of labeled macrophages was performed in pSNL/sham C57BL/6 mice at day 9 after surgery. Twenty-four h later,

anesthetized mice were put inside the pre-heated chamber. NIR imaging was performed before and 1 and 3 h after HC03 (100 mg kg$^{-1}$, i.p.) administration. Images were acquired with *Photo Acquisition* software and processed with M3 Vision software (Biospace Laboratory, Paris, France). The NIR pixel area was measured by ImageJ 1.32J from a region of the same size over the sciatic nerve, identifying an area of interest (ROI) around the fluorescent signal evident in the pSNL. That ROI perimeter was then reported to other images derived from different experimental settings. Data were expressed as NIR pixel area/ROI pixel area.

**Statistical analysis**. Statistical analysis was performed by the unpaired two-tailed Student's *t*-test for comparisons between two groups and the one-way or two-way ANOVA followed by the post hoc Bonferroni's test for comparisons of multiple groups. Sample sizes for experiments were determined based on previous studies, to ensure appropriate statistical power. $P < 0.05$ was considered statistically significant (GraphPad Prism version 5.00, La Jolla, USA).

**Data availability**. The data that support the findings of this study are available from the corresponding author on reasonable request.

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

## Acknowledgements

We thank A.H. Morice (University of Hull, UK) for the hTRPA1-HEK293 cells and D. Preti (University of Ferrara, Italy) for providing HC-030031. We also thank Mary K. Lokken for her expert English revision. This work was supported by grants from Istituto Toscano Tumori (ITT), grant 2014 and Regione Toscana, grant Nutraceuticals 2014, 'POFCADT' (to P. Geppetti).

## Author contributions

F.D.L., R.N., R.P., N.W.B. and P.G. designed experiments and interpreted results. F.D.L., R.N., S.M., M.C.G., I.M.M., D.R.D.I., J.F., S.L.P., S.B., G.T. and D.S.M.d.A. performed experiments. F.D.L., R.N., R.P., N.W.B. and P.G. wrote the manuscript.

## Additional information

**Competing interests:** The authors declare no competing financial interests.

