## [Peer Review File · Nature Communications]

Reviewers' comments:

Reviewer #1 (Remarks to the Author):

In this manuscript, De Logu, Nassini et al. show that macrophages recruited to injured peripheral nerves mediate prolonged neuropathic pain by producing NOX2-dependent oxidative stress, which activates TRPA1 on Schwann cells. This leads to: effective and sustained recruitment of macrophages to the endoneurium and thus prolonged macrophage-induced activation of Schwann cell TRPA1, which in turn activates Schwann cell NOX1 that produces oxidative stress leading to TRPA1 activation on nociceptors, and thus to sustained neuropathic pain. Overall, this study is very well conducted and presented with appropriate statistical analyses, complete, convincing and the findings represent important conceptual and mechanistic advances on how neuroinflammation sustains neuropathic pain after injury.

I have however a few comments below that the authors need to address before publication:

1/ For a Nature Communications manuscript, an introduction would help to better evaluate the background of the study and missing knowledge.

2/ Fig. 1a: At which time-point macrophage recruitment and oxidative stress were quantified? At 10 days? It would be useful to show a time-course of macrophage recruitment and oxidative stress to compare with allodynia (same time-points). Also, more information in the figure legends is necessary.

3/ Conclusion of 1st paragraph: Tone-down statement in the last sentence. It is in the present form too strong, because the data presented do not allow to make such a conclusion, and this would also help bringing smoothly the next part of the study that actually does not support the hypothesis that "macrophage-induced oxidative stress activate TRPA1 on nociceptors to mediate neuropathic pain". For example, "Together, these data potentially supported the hypothesis that macrophage-induced oxidative stress may activate TRPA1 on nociceptors to mediate neuropathic pain."

4/ Line 74: Only Fig. 1d-e (and not Fig. 1d-f) show that "Trpa1 deletion prevented infiltration of F4/80+ cells and H2O2 generation in the injured sciatic nerve". Please, correct.

5/ Line 79: The two first sentences of this paragraph make opposite statements without guiding the reader through. It is confusing. Please, adjust this. For example: "One possible explanation... may be that TRPA1... . However, as neither TRPA1 deletion/antagonism..."

6/ What did the authors use for efficient delivery of antisense oligonucleotides? Liposomes? Polymers? More information is needed on this point in the methods section. The authors also need to show the efficiency of oligonucleotides delivery with fluorescently labeled oligonucleotides or provide a reference that demonstrates efficient delivery using the same methods as they used.

Reviewer #2 (Remarks to the Author):

This study tested the hypothesis that TRPA1 in Schwann cells regulates neuropathic pain by causing macrophage infiltration through CCL2. I have serious concerns about the major conclusions of this study due to lack of specific tools. I also have concerns about

the novelty, the definition of neuroinflammation, and the role of CCL2.

1. This study lacks selective tool to address the role of Schwann cell TRPA1 in neuropathic pain. Intrathecal antisense oligodeoxynucleotides will affect *Trpa1* mRNA in the spinal cord (e.g. astrocytes and axons/terminals). Perineural antisense oligodeoxynucleotides may also affect *Trpa1* mRNA in nerve axons and non-Schwann cells in the nerve. Unfortunately, cell type specific deletion of *Trpa1* using conditional knockout mice or viral vectors was not considered in this study.

2. In many parts of this study, novelty is limited:

2a. Previous studies from this group have already shown TRPA1 activation by oxidative stress. This study does not reveal novel activators of TRPA1.

2b. Contribution of TRPA1 and oxidative stress to mechanical and cold allodynia is well known.

2c. It is well known that CCL2 can cause infiltration of macrophages for the induction of neuropathic pain.

2d. It is also known that CCL2 can sensitize TRPV1 in DRG neurons. It is not surprising that CCL2 can potentiate TRPA1 function in DRG neurons and axons.

3. Neuroinflammation is not clearly defined by the authors in this study.

a. Measurement of number of F4/80+ cells is just a small portion of neuroinflammation. Nerve injury also causes infiltration neutrophils and T cells as well as production of inflammatory mediators such as cytokines.

b. TRPA1 in nociceptors is known to cause neurogenic inflammation. CCL2 can be secreted from DRG neurons to cause neuroinflammation.

4. The actions of CCL2 are confusing. Many groups have shown that CCL2 is a neuromodulator and can directly sensitize DRG neurons and nerve axons via neuronal CCR2. This is supported by Fig. 2b, showing that perineural administration of CCL2 induced rapid mechanical allodynia. The slow actions of CCR2 antibody may be due to slow diffusion and binding of the antibody.

5. Fig. 3b. TRPA1 expression in the sciatic nerve is not very convincing. In addition to Schwann cells, nerve axons should also express TRPA1. Low magnification images are needed.

Reviewer #3 (Remarks to the Author):

De logu et al. presume a dual role of Schwann cell TRPA1 in neuropathic pain. They suggest that recruited macrophages at sites of injury via NOX2-dependent oxidative burst, trigger the Schwann cell TRPA1/NOX1 pathway, which generates a prolonged oxidative stress, which sustains macrophage infiltration within the injured nerve. TRPA1-dependent oxidative stress from Schwann cells also sends paracrine signals to TRPA1 of the ensheathed nociceptors to mediate allodynia.

It is an original idea and the MS is generally well written. However, some points of the manuscript necessitate further clarification. The major comments and questions are the following:

1. The novelty of the present manuscript is limited in some aspects, since the authors have recently published a similar study in the Brain. Trevisan et al. TRPA1 mediates trigeminal neuropathic pain in mice downstream of monocytes/macrophages and oxidative stress. (Brain. 2016 May;139(Pt 5):1361-77. doi: 10.1093/brain/aww038) , where they proposed that "trigeminal neuropathic pain, pain-like behaviours are entirely mediated by the TRPA1 channel, targeted by increased oxidative stress by-products released from monocytes and macrophages clumping at the site of nerve injury."
2. In the histological figures only the fluorescent pictures have been shown and the structure of the tissue barely recognizable. Have the authors also used general histological staining to identify macrophages besides F24/80?
3. Identification of Schwann cells have been based on S100 positivity, but there are some publications that it is not the best marker.(PLoS One. 2015; 10(4): e0123278. Specific Marker Expression and Cell State of Schwann Cells during Culture In Vitro)
4. It is amazing that the Abcam TRPA1 antibody binding is completely negative in the TRPA1 receptor KO mice. Which Abcam TRPA1 AB has been used in this study? Have the authors tested the AB on DRG neurons of WT and TRPA1 KO mice?
5. If the authors presume the cross-talk between Schwann-cell and neural TRPA 1 (as depicted in the last figure- graphical abstract) it would be important to do the double/triple staining of nerve fibres, TRPA1 and Schwann cells in the PNL model.
6. Have the authors measured the TRPA1 mRNA level in the cultures Schwann cells proving its synthesis by the glial cell? Is this possible to measure the TRPA1 receptor protein quantitatively by Western blotting in the Schwann cell culture?

Minor note: The structure of the manuscript is erratic. The reference list has been cut and it is in the middle of the text.

Reviewers' comments:

Reviewer #1 (Remarks to the Author):

In this manuscript, De Logu, Nassini et al. show that macrophages recruited to injured peripheral nerves mediate prolonged neuropathic pain by producing NOX2-dependent oxidative stress, which activates TRPA1 on Schwann cells. This leads to: effective and sustained recruitment of macrophages to the endoneurium and thus prolonged macrophage-induced activation of Schwann cell TRPA1, which in turn activates Schwann cell NOX1 that produces oxidative stress leading to TRPA1 activation on nociceptors, and thus to sustained neuropathic pain. Overall, this study is very well conducted and presented with appropriate statistical analyses, complete, convincing and the findings represent important conceptual and mechanistic advances on how neuroinflammation sustains neuropathic pain after injury.

I have however a few comments below that the authors need to address before publication:

1/ For a Nature Communications manuscript, an introduction would help to better evaluate the background of the study and missing knowledge.

We agree with this comment. As the manuscript was transferred directly from Nature to Nature Communications, no format change was possible at that stage. The current revised version complies with the Nature Communication format.

2/ Fig. 1a: At which time-point macrophage recruitment and oxidative stress were quantified? At 10 days? It would be useful to show a time-course of macrophage recruitment and oxidative stress to compare with allodynia (same time-points). Also, more information in the figure legends is necessary.

As the focus of the paper was to assess the role of TRPA1 when the allodynia is established (as usually happens in patients), macrophage infiltration and H₂O₂ generation were studied at day 10 after surgery. Nevertheless, the revised manuscript/figures/figure legends report additional information, including the time-course of macrophage infiltration and H₂O₂ generation at 3, 7, 10 and 20 days after surgery. Notably, the changes parallel the time-course of the mechanical allodynia (Figure 1b-d and page 5, line 3).

3/ Conclusion of 1st paragraph: Tone-down statement in the last sentence. It is in the present form too strong, because the data presented do not allow to make such a conclusion, and this would also help bringing smoothly the next part of the study that actually does not support the hypothesis that “macrophage-induced oxidative stress activate TRPA1 on nociceptors to mediate neuropathic pain”. For example, “Together, these data potentially supported the hypothesis that macrophage-induced oxidative stress may activate TRPA1 on nociceptors to mediate neuropathic pain.”

We thank the reviewer for the suggestion. In the revised ms (with more space allowed), as suggested, the sentence has been deleted and the previous hypothesis (Trevisan et al, Brain, 2016) considered with caution throughout the ms.

4/ Line 74: Only Fig. 1d-e (and not Fig. 1d-f) show that “Trpa1 deletion prevented infiltration of F4/80+ cells and H2O2 generation in the injured sciatic nerve”. Please, correct.

In the revised ms, the sentence has been corrected (page 6, line 3).

5/ Line 79: The two first sentences of this paragraph make opposite statements without guiding the reader through. It is confusing. Please, adjust this. For example: “One possible explanation... may be that TRPA1... . However, as neither TRPA1 deletion/antagonism...”

In the revised ms, the sentence has been corrected (page 6, line 11).

6/ What did the authors use for efficient delivery of antisense oligonucleotides? Liposomes? Polymers? More information is needed on this point in the methods section. The authors also need to show the efficiency of oligonucleotides delivery with fluorescently labeled oligonucleotides or provide a reference that demonstrates efficient delivery using the same methods as they used.

We used fully phosphorothioate-modified ODNs, which is reported to be more effective in the rate of uptake and more specific in binding the target sequence, generally increasing ODNs potency (Brown et al., J Biol Chem. 1994;269:26801-5). We confirmed the efficiency of the intrathecal and perineural antisense ODN delivery by measuring mRNA and protein expression (RT-qPCR and immunolabeling, respectively) in DRGs and Schwann cells and the acute nociceptive response to TRPA1 and TRPV1 agonists. Our results indicate that ODN produced selective knockdown of TRPA1 in either Schwann cells or DRG neurons. Therefore, we did not consider using fluorescent labeled ODNs to assess delivery. As requested, we now provide detailed information and related references in the method section (Kiguchi et al., European Journal of Pharmacology 2008, 592:87-92; Ma and Quirion, Journal of Neurochemistry, 2006, 98:180-192; Bonet et al., Neuropharmacology 2013;65:206-12) (Supplementary information, page 5, line 5). More importantly, new experiments with *Plp-CreERT;Trpa1* mice reinforce the conclusion obtained with ODN delivery (page 10, line 13).

Reviewer #2 (Remarks to the Author):

This study tested the hypothesis that TRPA1 in Schwann cells regulates neuropathic pain by causing macrophage infiltration through CCL2. I have serious concerns about the major conclusions of this study due to lack of specific tools. I also have concerns about the novelty, the definition of neuroinflammation,

and the role of CCL2.

1. This study lacks selective tool to address the role of Schwann cell TRPA1 in neuropathic pain. Intrathecal antisense oligodeoxynucleotides will affect *Trpa1* mRNA in the spinal cord (e.g. astrocytes and axons/terminals).

Perineural antisense oligodeoxynucleotides may also affect *Trpa1* mRNA in nerve axons and non-Schwann cells in the nerve. Unfortunately, cell type specific deletion of *Trpa1* using conditional knockout mice or viral vectors was not considered in this study.

We agree that this point raised by referee 2, and underlined by the editor, is a major issue. Accordingly, we have generated and studied mice with targeted deletion of *Trpa1* in Schwann cells. Mice with tamoxifen inducible Cre-mediated recombination system driven by the mouse *Plp1*, proteolipid protein (myelin) 1 promoter (B6.Cg-Tg(*Plp1-cre/ERT*)3Pop/J) were crossed with 129S-*Trpa1*^{tm2Kykw/J} loxP mice to generate Schwann cell/oligodendrocytes specific *Trpa1* knock-out mice. The newly generated mice were used in additional pSNL experiments, which are now included in the revised ms (page 10, line 13 and Fig. 8). We confirmed that these mice lack Schwann cell TRPA1, while neuronal expression is confirmed. The new data show that mechanical allodynia, macrophage infiltration and H₂O₂ production were all attenuated in mice with specific deletion of TRPA1 in Schwann cells. These new data are in agreement with our findings with TRPA1 knockdown in Schwann cells.

2. In many parts of this study, novelty is limited:

We report, for the first time, that TRPA1 is expressed in Schwann cells, where it generates oxidative stress that is required for the continuous recruitment of macrophages inside the injured nerve. These data overturn the traditional view that macrophage-induced oxidative stress contributes to pain by activating nociceptor TRPA1. Instead, our findings indicate that macrophages activate an intermediate mechanism represented by Schwann cell TRPA1. Hence, the role of TRPA1 in neuropathic pain is not limited to TRPA1 signaling in nociceptors, but rather requires TRPA1 signaling to Schwann cells (upstream to sensory neurons), where TRPA1 orchestrates neuroinflammation and pain. This finding is completely original.

2a. Previous studies from this group have already shown TRPA1 activation by oxidative stress. This study does not reveal novel activators of TRPA1.

2b. Contribution of TRPA1 and oxidative stress to mechanical and cold allodynia is well known.

We agree that TRPA1 activation by oxidative stress is well known. The novelty reported in the present paper is that Schwann cell TRPA1 generates oxidative stress required for maintaining neuropathic pain. Thus, TRPA1 is not only a sensor of oxidative stress, but, by amplifying the oxidant signal, it also sustains

macrophage accumulation and pain signaling. See also above the reply to point 2.

2c. It is well known that CCL2 can cause infiltration of macrophages for the induction of neuropathic pain.

2d. It is also known that CCL2 can sensitize TRPV1 in DRG neurons. It is not surprising that CCL2 can potentiate TRPA1 function in DRG neurons and axons.

We agree that it is known that CCL2 provides a key contribution to macrophage recruitment to the site of nerve injury, and that its prolonged inhibition attenuates both neuroinflammation and pain. However, for the first time, we show that CCL2 does not *per se* provoke allodynia, and that its reported ability to directly 'sensitize' nociceptors is not relevant in the pSNL model. The present data reveal that the ability of CCL2 to recruit macrophages into the injured nerve trunk and to promote allodynia is dependent on TRPA1 expressed in Schwann cells. We confirm that CCL2 is necessary to promote macrophage accumulation by acting in a wide spatial range (from the blood stream to proximity of the injured nerve). However, CCL2 is not sufficient for the final macrophage accumulation into the injured nerve (a response mediated by oxidative stress produced by TRPA1/NOX1 in Schwann cells) that is key for producing allodynia.

3. Neuroinflammation is not clearly defined by the authors in this study.

We were unable to fully define the two types of inflammation due to space constraints. The current revised ms addresses in more detail this point (page 12, line 6).

a. Measurement of number of F4/80+ cells is just a small portion of neuroinflammation. Nerve injury also causes infiltration neutrophils and T cells as well as production of inflammatory mediators such as cytokines.

Neutrophils contribute to the early phase of neuroinflammation/pain. However, their role is limited at the first few days after nerve injury, as indicated by previous studies (Perkins et al., Neuroscience, 2000;101:745-57) and now confirmed in the current report (page 5, line 9 from the bottom and Supplementary fig. 4d). In particular, we found that treatment with clodronate diminished allodynia, but left unchanged the neutrophil number within the injured nerve in pSNL mice (Supplementary fig. 4a,d). Similar considerations apply to lymphocytes (Kim and Moalem-Taylor, Brain Res 2011;1405:95-108, and Supplementary fig. 4a,d). A redundancy of cytokines and chemokines is probable after nerve injury. Since prolonged CCL2 inhibition abolished allodynia, further analysis of the contribution of individual cytokines and chemokines is beyond the scope of the present study.

b. TRPA1 in nociceptors is known to cause neurogenic inflammation. CCL2 can be secreted from DRG neurons to cause neuroinflammation.

To the best of our knowledge, there is no robust evidence that the short-lived and self-limiting neurogenic inflammation (mediated by SP and CGRP) contributes to the prolonged neuroinflammation that occurs after nerve injury (page 11, line 1 from the bottom). Experiments of defunctionalization of primary sensory neurons with resiniferatoxin fully support this view (page 7, line 9 from the bottom).

4. The actions of CCL2 are confusing. Many groups have shown that CCL2 is a neuromodulator and can directly sensitize DRG neurons and nerve axons via neuronal CCR2. This is supported by Fig. 2b, showing that perineural administration of CCL2 induced rapid mechanical allodynia. The slow actions of CCR2 antibody may be due to slow diffusion and binding of the antibody.

We thank the reviewer for this comment that prompted us to perform new experiments to clarify this issue. Clodronate treatment depleted macrophages and eliminated allodynia, without affecting CCL2 levels within the injured nerve (page 6, line 3 from the bottom and Supplementary fig. 4a,f). Furthermore, CCL2 administration failed to evoke allodynia in macrophage-depleted mice (page 6, line 5 from the bottom and Supplementary fig. 4e). It is unlikely that the slow diffusion of the CCL2-Ab delays its ability to affect pSNL-evoked allodynia. In fact, the CCL2-Ab, which within 1 hour suppressed CCL2 levels, failed to affect allodynia recorded up to 6 hours after the CCL2-Ab administration (page 7, line 8 and Fig. 2d). These observations indicate that in pSNL CCL2 ability to promote allodynia by a direct action on sensory nerves is minor. The results support the view that the pro-allodynic action of CCL2 is due to its ability to recruit macrophages from the blood stream close to the site of injury.

5. Fig. 3b. TRPA1 expression in the sciatic nerve is not very convincing. In addition to Schwann cells, nerve axons should also express TRPA1. Low magnification images are needed.

To address this issue, we performed additional immunofluorescence experiments with triple staining in sciatic nerve by using TRPA1, S100 and PGP9.5 antibodies. We also now include low magnification images (page 8, line 2 and Fig. 3b,c)

Reviewer #3 (Remarks to the Author):

De logu et al. presume a dual role of Schwann cell TRPA1 in neuropathic pain. They suggest that recruited macrophages at sites of injury via NOX2-dependent oxidative burst, trigger the Schwann cell TRPA1/NOX1 pathway, which generates a prolonged oxidative stress, which sustains macrophage infiltration within the injured nerve. TRPA1-dependent oxidative stress from Schwann cells also sends

paracrine signals to TRPA1 of the ensheathed nociceptors to mediate allodynia.

It is an original idea and the MS is generally well written. However, some points of the manuscript necessitate further clarification. The major comments and questions are the following:

1. The novelty of the present manuscript is limited in some aspects, since the authors have recently published a similar study in the Brain. Trevisan et al. TRPA1 mediates trigeminal neuropathic pain in mice downstream of monocytes/macrophages and oxidative stress. (Brain. 2016 May;139(Pt 5):1361-77. doi: 10.1093/brain/aww038) , where they proposed that “trigeminal neuropathic pain, pain-like behaviours are entirely mediated by the TRPA1 channel, targeted by increased oxidative stress by-products released from monocytes and macrophages clumping at the site of nerve injury.”

We initially expected (as reported in the paper by Trevisan et al, Brain 2016) that oxidative stress generated by infiltrating macrophages was responsible for the TRPA1 activation in nociceptors and the ensuing neuropathic pain. However, to our surprise, we found that TRPA1 genetic/pharmacological blockade attenuated macrophage accumulation and oxidative stress levels inside the injured nerve. New data with Schwann cell specific *Trpa1* knock-out mice confirmed these findings and highlight the role of Schwann cell TRPA1 in neuropathic pain. We consider these findings to be novel and provocative, for they are counter to the common belief that a macrophage-dependent oxidative burst directly targets TRPA1 in nociceptors.

2. In the histological figures only the fluorescent pictures have been shown and the structure of the tissue barely recognizable.

In the revised version of the manuscript we show brightfield histological (hematoxylin and eosin) staining, which allows identification of macrophages (page 5, line 3 and Supplementary fig. 1).

3. Identification of Schwann cells have been based on S100 positivity, but there are some publications that it is not the best marker.(PLoS One. 2015; 10(4): e0123278. Specific Marker Expression and Cell State of Schwann Cells during Culture In Vitro).

We thank the reviewer for this comment. We performed additional experiments (immunofluorescence and RT-qPCR in sciatic nerve tissue and cultured Schwann cells) using another Schwann cell marker (SOX10), which confirmed those obtained with S100 (page 8, line 2 and Fig 3a,f,h).

4. It is amazing that the Abcam TRPA1 antibody binding is completely negative in the TRPA1 receptor KO mice. Which Abcam TRPA1 AB has been used in this study? Have the authors tested the AB on DRG neurons of WT and TRPA1 KO mice?

We used a rabbit polyclonal antibody to detect TRPA1 (cat num# ab58844, Abcam, Cambridge). We observed that this antibody stained DRG neurons from wild-type mice but not from TRPA1 KO mice, indicating selectivity (Fig. 3d). Our results confirm previous studies with the same antibody (Brierley et al., *Gastroenterology* 2009, 137: 2084–2095) that indicate TRPA1 staining in nerve fibers of the colon of wild-type but not TRPA1 KO mice. Of note, we used mice (B6.129P-Trpa1tm1Kyk/J; Jackson Laboratories, Bar Harbor, ME, USA) originated by the same colony (heterozygotes on a C57BL/6 background) used by Brierley et al. (*Gastroenterology* 2009) and generated by Kelvin Kwan (Kwan et al., *Neuron* 2006;50:277-89).

5. If the authors presume the cross-talk between Schwann-cell and neural TRPA 1 (as depicted in the last figure- graphical abstract) it would be important to do the double/triple staining of nerve fibres, TRPA1 and Schwann cells in the PNL model.

We thank the reviewer for the suggestion. In the revised version of the manuscript we added images with a triple staining of nerve fibers (PGP9.5), TRPA1 and Schwann cells (S100) (page 8, line 2 and Figure 3c).

6. Have the authors measured the TRPA1 mRNA level in the cultures Schwann cells proving its synthesis by the glial cell?

The original ms reported representative RT-qPCR plot for S100 and TRPA1,

in cultured Schwann cells. The revised ms reports a quantitative analysis of the amount of TRPA1 mRNA in cultured Schwann cells (page 8, line 6 and Fig. 3h).

7. Is this possible to measure the TRPA1 receptor protein quantitatively by Western blotting in the Schwann cell culture?

Again, we thank the reviewer for the suggestion. New experiments allowed to quantify by Western blotting TRPA1 protein expression in Schwann cells (page 8, line 6 and Fig. 3g).

8. Minor note: The structure (of the manuscript is erratic. The reference list has been cut and it is in the middle of the text.

We again apologize. The awkward presentation of the ms was due the fact that the ms has been transferred automatically to Nat Comm from another journal (see above). We hope to have solved all these issues in the revised ms.

Reviewers' comments:

Reviewer #1 (Remarks to the Author):

The authors have appropriately answered my comments and brought further evidence of the contribution of Schwann cell TRPA1 to neuropathic pain with the inducible PlpCreERT-Trpa1 KO mice.

However, I have one remaining comment concerning the immunofluorescence (IF) images presented in Fig. 8b: Why does TRPA1 signal look that much different in the co-IF with S100 compared to the co-IF with PGP9.5? This is rather irritating. The authors need to keep in mind that recombination with Plp-CreERT mice is usually not 100%, thus it can be expected that a reduced number of Schwann cells will still express TRPA1. To appreciate the efficiency of recombination, co-IF of TRPA1 with S100 and with PGP9.5 in nerves of control versus KO animals is needed here. It is also necessary to precise at what time-point after tamoxifen injection animals were used for experiment. Was TRPA1 protein (partially) lost before carrying out pSNL?

Reviewer #2 (Remarks to the Author):

This revision is improved by including new studies on immunostaining and conditional deletion of TRPA1 in Schwann-like cells (Plp1+).

However, the authors still ignore the large literatures showing that CCL2 is a neuromodulator by regulating nociceptive neuron activity in DRG and spinal cord. CCL2 has direct effects on neurons. Intrathecal and intraplantar CCL2 can induce pain within an hour via neuronal modulation. After nerve injury, peri-neural injection of CCR2 antagonist should reduce pain within hours. The rapid effects of CCL2 and CCR2 are mediated by neurons, although the slow effects of CCL2/CCR2 involve both neuronal and immune cells.

Reviewer #3 (Remarks to the Author):

The author have made substantial improvement in the manuscript. The novel data on the Schwann cell specific Trpa1 knock-out mice increased the value of the study. It is amazing that the authors developed and provided a sufficient number of Schwann cell specific Trpa1 knock-out mice in 8 weeks. I have found satisfactory the answers and accept the replies for my concerns.

Reviewers' comments:

Reviewer #1 (Remarks to the Author):

The authors have appropriately answered my comments and brought further evidence of the contribution of Schwann cell TRPA1 to neuropathic pain with the inducible PlpCreERT-Trpa1 KO mice.

However, I have one remaining comment concerning the immunofluorescence (IF) images presented in Fig. 8b: Why does TRPA1 signal look that much different in the co-IF with S100 compared to the co-IF with PGP9.5? This is rather irritating. The authors need to keep in mind that recombination with Plp-CreERT mice is usually not 100%, thus it can be expected that a reduced number of Schwann cells will still express TRPA1. To appreciate the efficiency of recombination, co-IF of TRPA1 with S100 and with PGP9.5 in nerves of control versus KO animals is needed here. It is also necessary to precise at what time-point after tamoxifen injection animals were used for experiment. Was TRPA1 protein (partially) lost before carrying out pSNL?

The remarkable difference between co-IF of TRPA1 with S100 and co-IF of TRPA1 with PGP9.5 derived from our intention to show the most marked distinction. In other areas, not shown in the original manuscript, some co-staining between TRPA1 and S100 was detected. We have now produced additional images with triple S100, PGP9.5 and TRPA1 staining in *Plp1-Cre^{ERT};Trpa1^{fl/fl}* and control mice. We have generated a new, more representative image and semi-quantitative analysis (Figure 8b) that show a reduction, but not a complete elimination, of TRPA1 in S100⁺ Schwann cells of *Plp1-Cre^{ERT};Trpa1^{fl/fl}* mice. The text has been revised to reflect this change (page 10, lines 13 and 18).

We thank the referee for the second question. The time-point after tamoxifen when animals were used for experiments, which was not indicated in the original manuscript, is now reported in the method section (Supplementary information, page 3, line 4). pSNL or sham surgery was performed 15 days after the last injection of tamoxifen. At the same, time-point tissues/cells were collected to check the expression of TRPA1 in Schwann cells.

Reviewer #2 (Remarks to the Author):

This revision is improved by including new studies on immunostaining and conditional deletion of TRPA1 in Schwann-like cells (Plp1+).

However, the authors still ignore the large literatures showing that CCL2 is a neuromodulator by regulating nociceptive neuron activity in DRG and spinal cord. CCL2 has direct effects on neurons. Intrathecal and intraplantar CCL2 can induce pain within an hour via neuronal modulation. After nerve injury, peri-neural injection of CCR2 antagonist should reduce pain within hours. The rapid effects of CCL2 and CCR2 are mediated by neurons, although the slow effects of CCL2/CCR2 involve both neuronal and immune cells.

We thank the reviewer for these comments and agree with her/his view that CCL2 is an important neuromodulator. Indeed, our experiments confirmed the contribution of CCL2 action *via* inflammatory cells in the remarkably prolonged pain observed in the pSNL model. However, as the aim of our paper was to study the role of macrophages in the pSNL model, we did not investigate the direct and very rapid effect of CCL2 on DRG neurons to evoke neuronal sensitization and activation. Nevertheless, in the revised manuscript, the acute and direct role of CCL2 in pain has been better discussed, providing a new additional reference (White et al., *PNAS*, 2005) to the pre-existing four references on this topic (page 13, line 8).

Reviewer #3 (Remarks to the Author):

The authors have made substantial improvement in the manuscript. The novel data on the Schwann cell specific Trpa1 knock-out mice increased the value of the study. It is amazing that the authors developed and provided a sufficient number of Schwann cell specific Trpa1 knock-out mice in 8 weeks. I have found satisfactory the answers and accept the replies for my concerns.

We thank the reviewer for her/his appreciation of the satisfactory revision of our manuscript.

I wish to comment on the reviewer's amazement regarding the generation of the cell-specific TRPA1 knockout mice. In anticipation of requiring genetic evidence to support our studies on the role of Schwann cell TRPA1, we purchased from Jackson Laboratories the B6.Cg-Tg(Plp1-CreERT)3Pop/J (Charles River, Milan, Italy; invoice number EEC03260, May 04, 2016) and 129S-Trpa1^{tm2Kykw/J} mice (Charles River, Milan, Italy; invoice number EEC05339, July 27, 2016) one year ago. When we first submitted the manuscript (March 2017), the progeny of these mice was not available. We had anticipated that the AS-ODN-TRPA1 experiments were sufficient to support our hypothesis. However, within two months after the Nat Comm reviews, the mice were available for study.

REVIEWERS' COMMENTS:

Reviewer #1 (Remarks to the Author):

The authors have appropriately answered my comments and made the requested changes to the manuscript. This study is very interesting and the revised manuscript is to my view suitable for publication in Nature Communications.

Reviewer #2 (Remarks to the Author):

The authors have addressed my remaining concern.

Reviewer #3 (Remarks to the Author):

I suggest to accept the manuscript in the present form.